# Genome-wide association meta-analysis identifies five loci associated with postpartum hemorrhage

Bleeding in early pregnancy and postpartum hemorrhage (PPH) bear substantial risks, with the former closely associated with pregnancy loss and the latter being the foremost cause of maternal death, underscoring the severe impact on maternal–fetal health. We identified five genetic loci linked to PPH in a meta-analysis. Functional annotation analysis indicated candidate genes *HAND2*, *TBX3* and *RAP2C/FRMD7* at three loci and showed that at each locus, associated variants were located within binding sites for progesterone receptors. There were strong genetic correlations with birth weight, gestational duration and uterine fibroids. Bleeding in early pregnancy yielded no genome-wide association signals but showed strong genetic correlation with various human traits, suggesting a potentially complex, polygenic etiology. Our results suggest that PPH is related to progesterone signaling dysregulation, whereas early bleeding is a complex trait associated with underlying health and possibly socioeconomic status and may include genetic factors that have not yet been identified.

Pregnancy-associated bleeding can occur at all stages of pregnancy. Bleeding in early pregnancy can range from a benign event with no adverse effects to an indication of ongoing pregnancy loss and even serve as a potential marker for later pregnancy loss, obstetric complications and long-term maternal comorbidities[1,2]. Postpartum hemorrhage (PPH) is the leading cause of maternal mortality, with approximately 100,000 young and otherwise healthy women dying every year worldwide[3]. Despite affecting more than one in ten births and being a heritable condition, PPH remains unexplored at the genetic and molecular level[4]. Prior candidate gene studies have focused on genes involved in the coagulation pathways[5]. Even though the etiology of PPH is multifactorial, it often occurs even when established risk factors are not present[6,7].

The primary cause of PPH is uterine atony, which accounts for 70% of all cases[3]. Other causes include retained placental tissue, trauma and congenital or acquired coagulation disorders. Early identification and correct management of PPH can prevent maternal mortality and morbidity[8]. Therefore, there is great interest in assessing PPH risk before labor, and a growing body of literature has described detailed

prognostic models. However, a recent review showed that almost half of the existing prognostic models include features that can only be obtained postpartum[9]. Consequently, there is an urgent clinical need to understand the molecular etiology and identify biomarkers that characterize high-risk women before labor to initiate timely preventive measures and monitoring.

Here, we report the results of genome-wide association studies (GWAS) of up to 302,894 women from six Northern European cohorts to identify the genetic etiology of bleeding during different stages of pregnancy. Our results reveal complexity in the genetics of early bleeding and highlight the importance of the myometrium and progesterone-responsive genes in the etiology of PPH.

## Results

### Overall findings

Combining data from six Northern European cohorts including 331,792 women, we investigated the genetic architecture of three phenotypes related to bleeding during pregnancy: early bleeding (28,898 cases), antepartum bleeding (3,236 cases) and PPH (21,521 cases) (Fig. 1a).

✉e-mail: Soren.brunak@cpr.ku.dk; Henriette.svarre.nielsen@regionh.dk

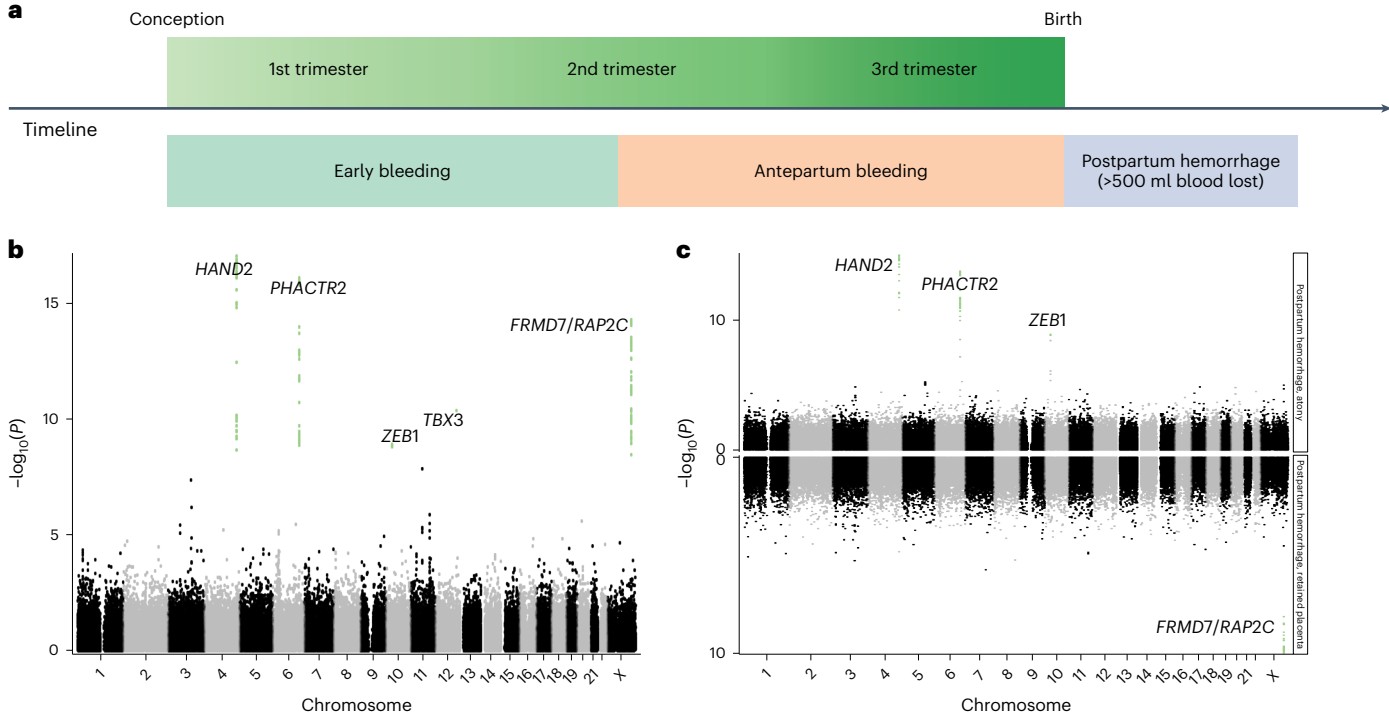

**Fig. 1 | Study overview and Manhattan plots. a**, Overview of the phenotypes under investigation. Early bleeding occurs up to and until the 20th gestational week, antepartum bleeding occurs between the 20th gestational week and birth, and PPH occurs after birth. **b**, Manhattan plot of PPH showing the 18 M variants, with SNPs passing the functionally informed multiple testing criteria highlighted in green and labels with gene names for the genes indicated by the functional analysis. **c**, Miami plot comparing PPH caused by atony (top) and retained placenta (bottom). Green dots indicate SNPs passing the multiple testing threshold, and labels hold the gene names for the genes indicated by the functional analysis.

Instances of overlap were observed, and the extent of such overlap exhibited variation across cohorts (Supplementary Fig. 1). We further divided early bleeding into 'early bleeding with any outcome' (28,898 cases) and 'early bleeding ending in live birth' (6,356 cases) (Supplementary Table 1). We included 18,009,056 sequence variants in a meta-analysis and identified five loci (chromosomes 4, 6, 10, 12 and X) that were associated with PPH using a functionally informed multiple testing correction (Fig. 1b and Table 1). The effect sizes were similar across all cohorts (Supplementary Fig. 2a), and conditional analysis revealed no secondary signals. We observed no significant associations between early bleeding and antepartum bleeding (Supplementary Figs. 3–5). In addition, we analyzed PPH as a result of uterine atony (13,048 cases and 261,809 controls) and PPH as a result of retained placental tissue (6,256 cases and 266,427 controls), in which three (chromosome 4, 6 and 10) and one (chromosome X) of the five associated loci passed multiple testing correction, respectively (Fig. 1c and Supplementary Table 2). We did not observe any significant differences in effect sizes between uterine atony and retained placental tissue when comparing the lead variants from the five loci (Supplementary Table 2). We found no evidence of confounding or inflation in any of the analyses (Supplementary Table 3).

### Prior evidence of single nucleotide polymorphisms
According to the GWAS catalog[10], the lead variant on chromosome 12 has previously been found in association with heel bone mineral density and prostate-specific antigen levels in males, both of which are hormone-responsive tissues. Additionally, the lead variants on chromosomes 10 and X were in strong ($r^2 > 0.8$) linkage disequilibrium (LD) with variants associated with uterine fibroids and endometriosis, while the lead variant on chromosome 6 was in strong LD with a sequence variant associated with educational attainment (Supplementary Table 4). Furthermore, we investigated the genome-wide

significant lead variants in the FinnGen cohort (release 9) and found that the lead variants on chromosome 12 (*TBX3*) and chromosome X (*FRDM7/RAP2C*) were also associated with endometriosis, and the loci on chromosomes 6 (*PHACTR2*), 10 (*ZEB1*) and X (*FRDM7/RAP2C*) were associated with uterine fibroids (Table 1 and Supplementary Fig. 2b).

### In silico functional analysis of loci
We annotated the five PPH lead variants and their correlated variants ($r^2 > 0.80$), hereafter referred to as PPH signals, according to their location in the ENCODE encyclopedia of candidate *cis*-regulatory elements (cCREs)[11]. Collectively, cCREs span 291 Mb of the genome and contain 10.2% of sequence variants. We found that all five PPH signals were located within either the distal or proximal enhancer-like sequences, suggesting non-coding regulatory functions (Supplementary Tables 5–7).

The predicted gene targets for these regulatory elements in uterine tissue are *TBX3* (12q24.21), *FRMD7* and *RAP2C* (Xq26.2) according to Epimap[12] (Supplementary Tables 7–9). Furthermore, there is evidence that the lead single nucleotide polymorphism (SNP) at the chromosome 4 locus, rs13141656, targets *HAND2* in endometrial tissue[13,14]. None of these genes have been directly associated with PPH. *HAND2* and *TBX3* are involved in stromal–epithelial communication during implantation. *HAND2* is implicated in preterm birth and gestational duration and has previously been found to be critical for implantation[15,16]. The function of the *RAP2C/FRMD7* gene cluster is currently unknown, but variants in the *RAP2C* locus are associated with gestational duration[17]. None of the proteins are known to physically interact according to the STRING database (v.11.5)[18].

We tested the PPH signals for enrichment within 1,210 transcription factor binding sites in DNA of various cell types and tissues[19], amounting to a total of 4,143 tests, and we used Bonferroni correction to set the threshold for significances at $P < 0.05 / 4,143$ ($-1 \times 10^{-5}$).

**Table 1 | Effect sizes across loci for PPH, endometriosis and uterine fibroids**

| CHR | BP (hg38) | RSID | Effect allele | Other allele | Effect allele Frequency | Odds ratio (95% CI, Pvalue) | | |
|---|---|---|---|---|---|---|---|---|
| | | | | | | PPH | Endometriosis | Uterine fibroids |
| 4 | 173807552 | rs13141656 | T | C | 0.30 | 1.10 (1.08–1.13; $1.42×10^{-17}$) | 0.98 (0.96–1.0; 0.014) | 0.98 (0.96–0.99; 0.00076) |
| 6 | 143642758 | rs12195857 | A | G | 0.32 | 1.10 (1.08–1.13, $9.86×10^{-17}$) | 0.97 (0.95–0.99; 0.0022) | 0.97 (0.96–0.98; $2.8×10^{-5}$) |
| 10 | 31660483 | rs11591307 | A | G | 0.22 | 1.08 (1.05–1.11, $1.3×10^{-9}$) | 1.03 (1.03–1.05; 0.015) | 0.94 (0.93–0.95; $4.6×10^{-16}$) |
| 12 | 114656455 | rs11067228 | G | A | 0.42 | 1.07 (1.05–1.10, $4.33×10^{-11}$) | 0.96 (0.95–0.98; $2.8×10^{-5}$) | 1.00 (0.99–1.02; 0.74) |
| X | 132131995 | rs2747025 | A | G | 0.32 | 0.91 (0.89–0.94, $9×10^{-15}$) | 0.93 (0.91–0.95; $1.2×10^{-14}$) | 1.17 (1.15–1.18; $4.6×10^{-113}$) |

Endometriosis and uterine fibroid estimates come from the datasets listed in Supplementary Table 4.

The number of PPH signals found in the progesterone receptor binding sites in human embryonic stem cells was significantly higher than expected ($P = 5 × 10^{-6}$; Table 2). Progesterone is an important factor in the establishment and maintenance of pregnancy and is therefore relevant in the context of PPH.

We used MAGMA[20] to test for tissue-specific enrichment using expression data from the Human Protein Atlas bulk tissue and single-cell datasets[21]. We found that the endometrium, smooth muscle, seminal vesicle and thyroid gland tissue were enriched, as well as endothelial cells (false discovery rate of <5%) (Fig. 2a,b).

### Maternal and fetal transmission

We performed a haplotype-specific analysis of the five PPH-associated variants in the MoBa and deCODE cohorts to distinguish between maternal and fetal effects. These results were consistent with all five variants affecting the risk of PPH primarily through the maternal genome (Supplementary Fig. 6 and Supplementary Table 10). However, we cannot exclude any effect from the fetal genome.

### Heritability of pregnancy-associated bleeding traits

We estimated the SNP heritability of early bleeding in pregnancy and PPH to be 12.7% (95% CI, 7.8–17.6%) and 16.5% (95% CI, 10.2–22.8%), respectively in the Danish cohort, assuming a population prevalence of 25% and 15%, respectively. We selected prevalences based on a literature review[2,8].

We characterized the intra-phenotypic genetic correlations among the five bleeding-in-pregnancy phenotypes investigated in this study (early bleeding in pregnancy, any outcome; early bleeding in pregnancy, live birth; PPH; PPH caused by atony; and PPH caused by retained placenta). Antepartum bleeding did not have a sufficient polygenic signal to be investigated (LD score, $χ^2 < 1.02$). Early bleeding during pregnancy did not exhibit any significant genetic correlation with PPH or any of its subtypes (Fig. 3a). Notably, there was a strong genetic correlation between PPH caused by uterine atony and PPH caused by retained placenta ($r_g = 0.77$, 95% CI, 0.49–1.05).

Next, we aimed to characterize the genetic overlap of early bleeding (any outcome) and PPH with other co-occurring diseases and other phenotypes. The range of phenotypes that may co-occur with early bleeding during pregnancy and PPH has not been extensively characterized. Consequently, we looked for associations in three distinct cohorts: the Estonian Biobank ($n = 17,094$), UK Biobank ($n = 12,490$) and a Danish nationwide cohort ($n = 2,320,776$). Following a meta-analysis of 417 and 628 ICD-10 codes at the third level for early bleeding and PPH, respectively, we found that 120 codes were significantly associated with PPH (false discovery rate of <0.05) and 625 codes were significantly associated with early bleeding (Supplementary File 1).

Based on the literature, known risk factors, lifestyle, socioeconomic factors and the pairwise phenotype-to-phenotype correlation analyses presented here, we identified a list of phenotypes for which we could find suitable summary statistics (Supplementary Table 11). We additionally included socioeconomic and cardiometabolic traits,

**Table 2 | PPH signals were enriched (P < 0.05, Bonferroni corrected) within binding sites for progesterone receptor (PGR) defined in human embryonic stem cells (hESC)**

| DNA binding protein | Tissue or cell line | Annotated PPH signals, n | Expected proportion of annotated PPH signals (%) | Pvalue |
|---|---|---|---|---|
| PGR | hESC | 4/5 | 5 | $5×10^{-6}$ |
| ZNF558 | HEK293 | 4/5 | 27 | $7×10^{-5}$ |
| PGR | Myometrium | 5/5 | 17 | 0.001 |
| PGR | Leiomyoma | 3/5 | 8.6 | 0.002 |
| IRF2BP2 | HEK293 | 3/5 | 9.9 | 0.005 |
| MED12 | Leiomyoma | 3/5 | 11 | 0.007 |
| MYOG | RH4 | 3/5 | 11 | 0.009 |
| MED12 | Myometrium | 3/5 | 14 | 0.01 |
| ONECUT1 | Hep-G2 | 3/5 | 15 | 0.019 |
| FOXA1 | Prostate | 4/5 | 17 | 0.02 |
| ZNF3 | Hep-G2 | 3/5 | 7 | 0.023 |

Shown are nominally significant results; that is, where uncorrected $P < 0.05$.
We defined binding sites by ChIP–seq data available through the Remap2022 database (remap.univ-amu.fr).

such as body mass index, smoking and blood pressure. These traits are not recorded in the registries but are highly correlated with the diseases we found in the phenotype-to-phenotype correlation analysis. PPH was, at the genetic level, strongly positively correlated with birth weight (maternal and fetal) and gestational duration (maternal) and had an inverse correlation with uterine fibroids (Bonferroni-corrected $P < 0.05$) (Fig. 3b; see Supplementary Table 11 for a description of the summary stats). No other traits displayed a significant genetic correlation with PPH after multiple testing corrections. Although no sequence variants were found in association with early bleeding, we nonetheless found genetic correlations to reproductive, socioeconomic, cardiovascular and psychiatric traits (Fig. 3c).

### Polygenic risk scores

Using 25,118 pregnancies ($n = 19,026$ women) since 2012 from the Danish cohort, we found that a logistic regression model including the polygenic risk score (PRS) for PPH and birth weight, the latter derived from the maternal genome, yielded an improved model (likelihood ratio test, $P < 2 × 10^{-16}$; Supplementary Table 12) compared to a model that included only age, pre-pregnancy body mass index, parity, prior number of cesarean sections and prior number of PPHs. The variance explained (Nagelkerke $R^2$) increased from 3.2% (2.7%; 3.8%, 95% percentile bootstrap interval) to 3.8% (3.4%; 4.5%, 95% percentile bootstrap interval), yielding a net improvement of 0.7% (0.5%; 0.9% 95% percentile bootstrap interval). A model that included only the PRS for PPH explained 3.4% (2.9%; 3.9%, 95% percentile bootstrap interval),

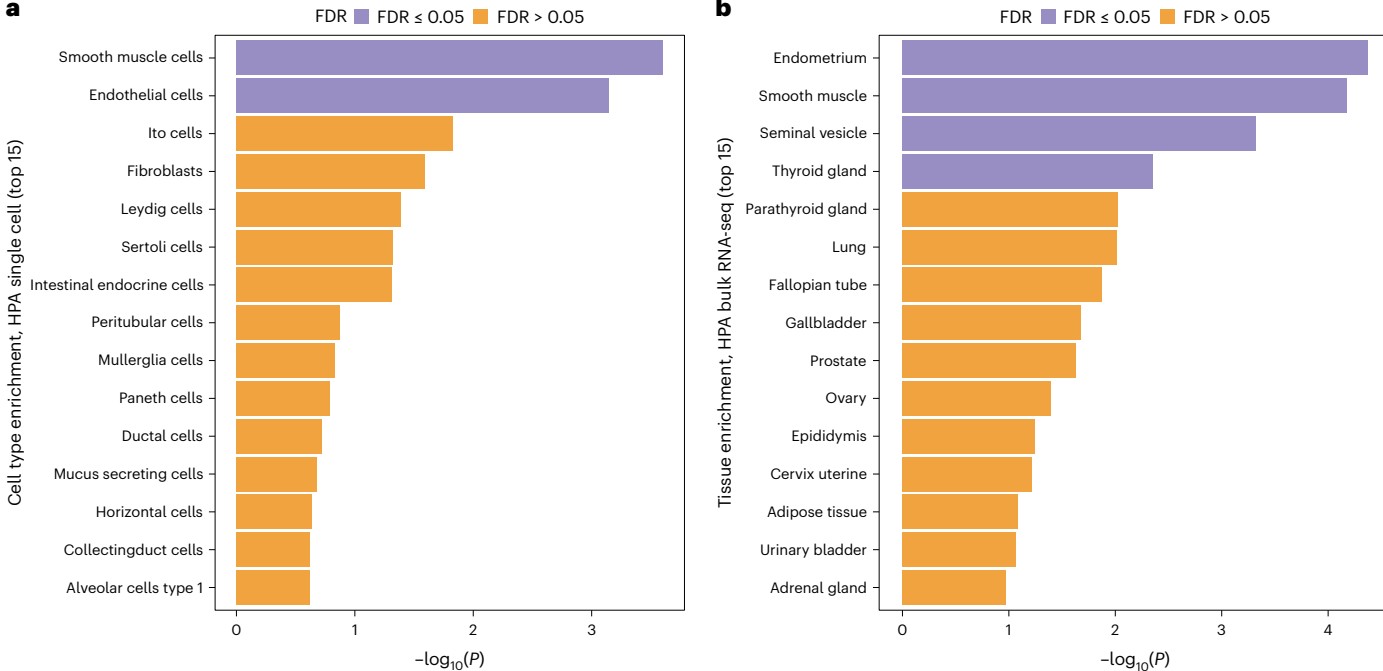

**Fig. 2 | Tissue-specific enrichment analysis for PPH. a**, MAGMA single-cell enrichment from the Human Protein Atlas (HPA). Smooth muscle cells and endothelial cells were both enriched (false discovery rate, FDR < 0.05).

**b**, MAGMA bulk tissue enrichment from the Human Protein Atlas showed enrichment of endometrial, smooth muscle, seminal vesicle and thyroid gland tissue (FDR < 0.05).

yielding an improvement of 0.2% (0.1%; 0.4%, 95% percentile bootstrap interval). Similarly, the area under the curve score increased from 0.60 (0.59; 0.61) to 0.61 (0.60; 0.62), improving marginally (0.008, 0.005; 0.011) when including both PRS. Including only the PRS for PPH resulted in a smaller improvement in the area under the curve value (0.003, 0.001; 0.006).

## Discussion

In this study, we investigated the genetic architecture of bleeding associated with pregnancy, which is one of the most common complications of pregnancy associated with both maternal and fetal morbidity and mortality. We identified five loci associated with PPH, with strong functional evidence of association with genes involved in implantation and contraction. Furthermore, enrichment of progesterone receptor binding sites substantiates the importance of hormone regulation in the etiology of PPH and suggests organ-specific dysregulation. However, in the absence of relevant tissue (myometrium sampled during or right before pregnancy), we were not able to locate the point or points in pregnancy at which the sequence variants exert their effect. There was no evidence of a genetic correlation between PPH and diseases. Our study revealed that early bleeding is highly polygenic with genetic correlations spanning various categories of human traits, and PPH is a disorder of hormone-responsive genes. Overall, this study provides insights into the genetic basis of bleeding during pregnancy and suggests different genetic pathways for early bleeding and PPH.

We analyzed data from six Northern European cohorts, representing six different countries with similar, albeit varying, universal healthcare systems, protocols for pregnancy care and levels of available clinical information. However, it is important to note that PPH disproportionately affects women in developing countries, and further research is needed to integrate more diverse populations into studies of this kind. Additionally, the registration of early bleeding during pregnancy depends heavily on the healthcare-seeking behavior of the individual and organization of early pregnancy care and is most likely affected by the heterogeneous causes of early bleeding. Not all

cohorts had information on early bleeding during pregnancy, and only three cohorts could distinguish between events leading to live births and those that did not. Another factor that should be considered is that oxytocin, a drug used to prevent or treat PPH, is administered pre-emptively based on other factors, such as cesarean section and PPH in a previous pregnancy. This bias most likely results in a smaller effect, thereby requiring increased sample sizes for the detection of associated loci.

The potential causal genes at the five loci that may contribute to the development of PPH were not related to previously suggested causes, such as the oxytocin receptor or coagulation cascade[5,22]; the latter being expected, as women with known coagulation disorders were excluded. The identified loci were found to be significantly enriched with progesterone-binding sites in human embryonic stem cells and showed nominal significance in the myometrium, the smooth muscle layer of the uterus responsible for contractions during labor and delivery. Progesterone is known to relax the myometrium and reduce contractility[23], which is vital for maintaining a healthy pregnancy. The presence of progesterone-binding sites suggests that genes in these regions, such as *HAND2*, which influences uterine development, *PHACTR2*, involved in actin regulation for muscle contraction, *ZEB1*, pivotal in tissue remodeling, *TBX3*, associated with developmental processes in uterine function and *RAP2C*, a mediator of cellular dynamics, may have roles in regulating myometrial contractility. Their biological functions and expression in uterine tissues substantiate their potential involvement in abnormal contractions that can lead to PPH. Furthermore, these loci were also associated with endometriosis and/or uterine fibroids. Endometriosis and uterine fibroids are both treated with selective progesterone receptor modulators, which target the progesterone receptor[24]. Nonetheless, owing to the lack of relevant in situ tissue, the exact timing of these genes is unknown. PPH was genetically correlated with birth weight and gestational duration. Fetal macrosomia and multiple gestations are believed to dilate the myometrial muscles of the uterus, making it more difficult to contract[25]. The correlation between gestational duration and PPH

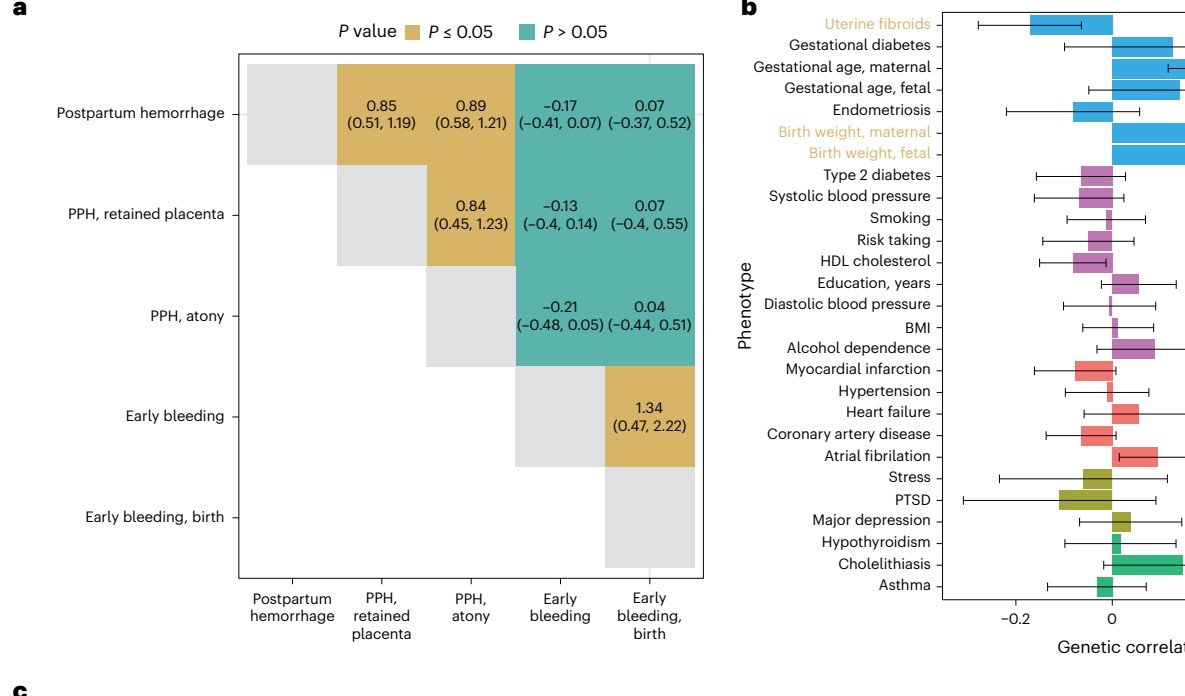

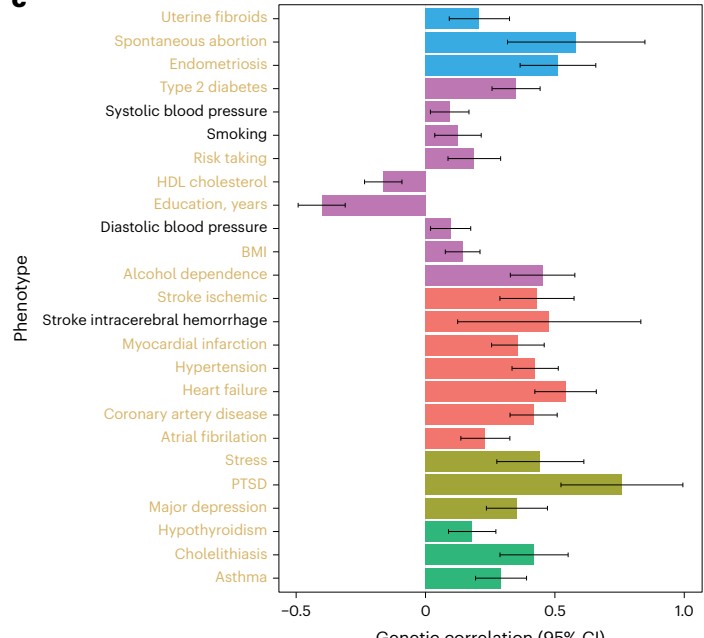

**Fig. 3 | Genetic correlation analysis. a,** Cross-trait genetic correlation of all bleeding in pregnancy phenotypes (with 95% confidence intervals). PPH and early bleeding in pregnancy show no noteworthy genetic correlation. PPH caused by atony or retained placenta are genetically indistinguishable. **b,** Genetic correlations between PPH and selected disorders. **c,** Genetic correlations between early bleeding and selected traits. Correlations that are significant after accounting for the number of traits tested are highlighted in yellow text. Error bars, 95% CI. The datasets used for the analysis are described in Supplementary Table 11. HDL, high-density lipoprotein; BMI, body mass index; PTSD, post-traumatic stress disorder.

is more complex, as it can be caused both by biological issues (uterine overdistension and placental issues) and medically induced labor[26]. Observational studies suggest that early bleeding, antepartum bleeding and PPH are correlated[2,27]. However, we did not observe any evidence of a shared genetic etiology. The effect of the identified loci was mediated primarily through the maternal genome. This is in line with a prior observational study that could not detect any fetal contribution to the heritability of severe PPH (>1,000 ml)[4]. Nonetheless, we cannot rule out fetal effects completely, as the cohorts with fetal genetic data available were not well powered.

We established early bleeding as a complex trait, substantiated by significant heritability, polygenic signals and widespread pleiotropy across disease areas. Early bleeding is related to pregnancy loss and may be an indication of the maternal body not coping well with the pregnancy. Genetic correlation with post-traumatic stress disorder and a variety of seemingly unrelated diseases and traits may be an indication of an extreme response to stress and a general low tolerance of the added burden of pregnancy upon maternal systems with underlying weaknesses. Early bleeding is a phenotype with high heterogeneity and may be a result of implantation bleeding, trauma, pregnancy loss,

abnormal products of conception (ectopic pregnancy, molar pregnancy), infections (pelvic inflammatory diseases, sexually transmitted diseases or bacterial vaginosis), cervical changes, subchorionic hemorrhage or unexplained causes. We did not identify any variants associated with early bleeding; therefore, we could not test for causality using, for example, Mendelian randomization. Nonetheless, a previous study indicated a causal relationship between early bleeding and cardiometabolic diseases[1].

The use of PRSs resulted in marginal improvements in the predictive capability for PPH. Nonetheless, as genetic studies become better powered, we can expect an improvement in their predictive capability. Consequently, the addition of PRSs to prognostic models should be considered in future studies to enable early stratification of women at a high risk of PPH.

Our findings reveal complex genetics of early bleeding in pregnancy. They further provide valuable insights into the potential underlying mechanisms of PPH and may inform the development of more effective prevention strategies.

## Online content

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

David Westergaard [1,2,3], Valgerdur Steinthorsdottir [4], Lilja Stefansdottir[4], Palle Duun Rohde [5], Xiaoping Wu [6,7], Frank Geller [6,7], Jaakko Tyrmi[8], Aki S. Havulinna[9,10], Pol Solé-Navais [11], Christopher Flatley [11], Sisse Rye Ostrowski [6,12], Ole Birger Pedersen [12,13], Christian Erikstrup [14,15], Erik Sørensen[6], Christina Mikkelsen [6,16], Mie Topholm Bruun [17,18], Bitten Aagaard Jensen[19], Thorsten Brodersen [13], Henrik Ullum[7], FinnGen*, Danish Blood Donor Study Genomic Consortium*, Estonian Biobank Research Team*, Nordic Collaboration for Womens and Reproductive Health*, Per Magnus [20], Ole A. Andreassen [21,22,23], Pål R. Njolstad [24,25], Astrid Marie Kolte[1], Lone Krebs [1,12], Mette Nyegaard [5], Thomas Folkmann Hansen [2,26], Bjarke Feenstra [6,7], Mark Daly[9,27,28], Cecilia M. Lindgren[28,29,30,31], Gudmar Thorleifsson [4], Olafur A. Stefansson [4], Gardar Sveinbjornsson [4], Daniel F. Gudbjartsson [4,32], Unnur Thorsteinsdottir[4,33], Karina Banasik[1,2], Bo Jacobsson[11,20], Triin Laisk [34], Hannele Laivuori [8,9,35,36], Kari Stefansson[4,33], Søren Brunak [2] ✉ & Henriette Svarre Nielsen [1,12] ✉

[1]Department of Obstetrics and Gynaecology, Copenhagen University Hospital Hvidovre, Hvidovre, Denmark. [2]Novo Nordisk Foundation Center for Protein Research, Faculty of Health and Medical Sciences, University of Copenhagen, Copenhagen, Denmark. [3]Methods and Analysis, Statistics Denmark, Copenhagen, Denmark. [4]deCODE genetics/Amgen, Reykjavik, Iceland. [5]Department of Health Science and Technology, Aalborg University, Gistrup, Denmark. [6]Department of Clinical immunology, Copenhagen University Hospital Rigshospitalet, Copenhagen, Denmark. [7]Department of Epidemiology Research, Statens Serum Institut, Copenhagen, Denmark. [8]Centre for Child, Adolescent, and Maternal Health Research, Faculty of Medicine and Health Technology, Tampere University, Tampere, Finland. [9]Institute for Molecular Medicine Finland, Helsinki Institute of Life Science, University of Helsinki, Helsinki, Finland. [10]Finnish Institute for Health and Welfare – THL, Helsinki, Finland. [11]Department of Obstetrics and Gynaecology, Institute of Clinical Sciences, Sahlgrenska Academy, University of Gothenburg, Gothenburg, Sweden. [12]Department of Clinical medicine, Faculty of Health and Medical Sciences, University of Copenhagen, Copenhagen, Denmark. [13]Department of Clinical immunology, Zealand University Hospital, Køge, Denmark. [14]Department of Clinical Immunology, Aarhus University Hospital, Aarhus, Denmark. [15]Department of Clinical Medicine, Aarhus University, Aarhus, Denmark. [16]Novo Nordisk Foundation Center for Basic Metabolic Research, Faculty of Health and Medical Science, University of Copenhagen, Copenhagen, Denmark. [17]Clinical Immunological Research Unit, Department of Clinical Immunology, Odense University Hospital, Odense, Denmark. [18]Department of Clinical Research, University of Southern Denmark, Odense, Denmark. [19]Department of Clinical Immunology, Aalborg University Hospital, Aalborg, Denmark. [20]Department of Genetics and Bioinformatics, Health Data and Digitalization, Norwegian Institute of Public Health, Oslo, Norway. [21]NORMENT Centre, University of Oslo, Oslo, Norway. [22]Institute of Clinical Medicine, Faculty of Medicine, University of Oslo, Oslo, Norway. [23]Division of Mental Health and Addiction, Oslo University Hospital, Oslo, Norway. [24]Mohn Center for Diabetes Precision Medicine, Department of Clinical Science, University of Bergen, Bergen, Norway. [25]Children and Youth Clinic, Haukeland University Hospital, Bergen, Norway. [26]Danish Headache Center, Department of neurology, Copenhagen University Hospital, Glostrup, Denmark. [27]Analytic and Translational Genetics Unit, Massachusetts General Hospital, Boston, MA, USA. [28]Broad Institute of MIT and Harvard, Cambridge, MA, USA. [29]Big Data Institute Li Ka Shing Centre for Health Information and Discovery, University of Oxford, Oxford, UK. [30]Nuffield Department of Population Health, University of Oxford, Oxford, UK. [31]Wellcome Trust Centre Human Genetics, University of Oxford, Oxford, UK. [32]School of Science and Engineering, Reykjavik University, Reykjavik, Iceland. [33]Faculty of Medicine, School of Health Sciences, Reykjavik University, Reykjavik, Iceland. [34]Estonian Genome Centre, Institute of Genomics, University of Tartu, Tartu, Estonia. [35]Medical and Clinical Genetics, University of Helsinki and Helsinki University Hospital, Helsinki, Finland. [36]Department of Obstetrics and Gynaecology, Tampere University Hospital, Tampere, Finland. *A list of members and their affiliations appears in the Supplementary Information. ✉e-mail: Soren.brunak@cpr.ku.dk; Henriette.svarre.nielsen@regionh.dk

## FinnGen

Jaakko Tyrmi[8], Aki S. Havulinna[9,10] & Hannele Laivuori[8,9,35,36]

## Danish Blood Donor Study Genomic Consortium

David Westergaard[1,2,3], Palle Duun Rohde[5], Frank Geller[6,7], Sisse Rye Ostrowski[6,12], Ole Birger Pedersen[12,13], Christian Erikstrup[14,15], Erik Sørensen[6], Christina Mikkelsen[6,16], Mie Topholm Bruun[17,18], Bitten Aagaard Jensen[19], Thorsten Brodersen[13], Henrik Ullum[7], Mette Nyegaard[5], Thomas Folkmann Hansen[2,26], Bjarke Feenstra[6,7], Karina Banasik[1,2], Kari Stefansson[4,33] & Søren Brunak[2]

## Estonian Biobank Research Team

Triin Laisk[34]

## Nordic Collaboration for Womens and Reproductive Health

David Westergaard[1,2,3], Valgerdur Steinthorsdottir[4], Palle Duun Rohde[5], Jaakko Tyrmi[8], Aki S. Havulinna[9,10], Pol Solé-Navais[11], Christopher Flatley[11], Mette Nyegaard[5], Mark Daly[9,27,28], Cecilia M. Lindgren[28,29,30,31], Unnur Thorsteinsdottir[4,33], Karina Banasik[1,2], Bo Jacobsson[11,20], Triin Laisk[34], Hannele Laivuori[8,9,35,36], Kari Stefansson[4,33], Søren Brunak[2] & Henriette Svarre Nielsen[1,12]

## Methods

### Study cohorts

This was a multi-national study that included six cohorts of Western European ancestry: the Copenhagen Hospital Biobank Study on Reproduction (Denmark), Estonian Biobank (Estonia), FinnGen (Finland), deCODE genetics (Iceland), UK Biobank (England) and Norwegian Mother, Father and Child Cohort Study (Norway). All studies were approved by the relevant institutional ethics review boards.

**Copenhagen Hospital Biobank Study on Reproduction and the Danish Blood Donor Study.** The Copenhagen Hospital Biobank (CHB) is based on EDTA blood samples collected from patients for blood typing and red cell antibody screening at hospitals in the Greater Copenhagen Area[28]. The CHB Study on Reproduction cohort focuses on patients with fertility and obstetric complications, identified through the Danish National Patient Registry. We also included blood donors from the Danish Blood Donor Study Genomic Cohort[29]. All samples were genotyped at deCODE genetics using the Illumina Infinium Global Screening array. Samples were imputed using an in-house pan-Scandinavian reference panel[30]. Association analysis was performed using software developed at deCODE genetics[31]. Approval of the CHB Reproductive Health Study was obtained from the Danish National Committee on Health Research Ethics (NVK-1805807) and the Capital Region Data Protection Agency (P-2019-49).

**Estonian Biobank.** The Estonian Biobank (EstBB) is a population-based biobank with >200,000 participants (~20% of the total Estonian population)[32,33]. In brief, all EstBB participants were genotyped using Illumina arrays at the Core Genotyping Lab of the Institute of Genomics, University of Tartu. Samples were imputed using a population-specific imputation reference of 2,297 whole-genome sequencing samples[34]. Association analysis was performed using SAIGE v.0.43.1. The activities of the EstBB are regulated by the Human Genes Research Act, which was adopted in 2000 specifically for the operations of the EstBB. All EstBB participants have signed a broad informed consent form, and analyses were carried out under ethical approval 1.1-12/624 from the Estonian Committee on Bioethics and Human Research (Estonian Ministry of Social Affairs) and data release N05 from the EstBB.

**FinnGen.** FinnGen is a public–private partnership research project that combines imputed genotype data generated from newly collected and legacy samples from Finnish biobanks and digital health record data from Finnish health registries (https://www.finngen.fi/en) with the aim of providing new insights into disease genetics[35]. FinnGen includes nine Finnish biobanks, research institutes, universities and university hospitals, 13 international pharmaceutical industry partners and the Finnish Biobank Cooperative in a pre-competitive partnership. As of November 2022 (release 10 described in this article), 412,181 individuals have been analyzed. The project uses data from the nationwide longitudinal health registers collected since 1969 from every resident in Finland. Participants in FinnGen provided informed consent for biobank research on the basis of the Finnish Biobank Act. Alternatively, separate research cohorts that were collected before the Finnish Biobank Act came into effect (in September 2013) and the start of FinnGen (August 2017) were compiled based on study-specific consent and later transferred to the Finnish biobanks after approval by Fimea, the National Supervisory Authority for Welfare and Health. Recruitment protocols followed the biobank protocols approved by Fimea. The Coordinating Ethics Committee of the Hospital District of Helsinki and Uusimaa approved the FinnGen study (protocol number HUS/990/2017). The FinnGen study is approved by the Finnish Institute for Health and Welfare (approval number THL/2031/6.02.00/2017,

amendments THL/1101/5.05.00/2017, THL/341/6.02.00/2018, THL/2222/6.02.00/2018, THL/283/6.02.00/2019 and THL/1721/5.05.00/2019), the Digital and Population Data Service Agency (VRK43431/2017-3, VRK/6909/2018-3 and VRK/4415/2019-3), the Social Insurance Institution (KELA) (KELA 58/522/2017, KELA 131/522/2018, KELA 70/522/2019 and KELA 98/522/2019) and Statistics Finland (TK-53-1041-17).

**deCODE genetics.** The deCODE cohort is a nationwide sample collection project that has been ongoing in Iceland since 1997. All participants who donated blood signed an informed consent form. Variants were identified through whole-genome sequencing of 63,460 individuals. They were imputed into 173,025 chip-genotyped Icelanders using long-range phasing, and into their untyped close relatives based on genealogy[31,36]. We used logistic regression to test for association of sequence variants assuming an additive genetic model, using software developed at deCODE genetics[31]. The deCODE study was approved by the Icelandic National Bioethics Committee (VSN-15-169).

**Norwegian Mother, Father and Child Cohort Study.** The Norwegian Mother, Father and Child Cohort Study (MoBa) is a population-based pregnancy cohort study conducted by the Norwegian Institute of Public Health. Participants were recruited from all over Norway from 1999 to 2008 (ref. 37). The women consented to participation in 41% of the pregnancies. The cohort includes approximately 114,500 children, 95,200 mothers and 75,200 fathers. The current study is based on version 12 of the quality-assured data files released for research. Details about PPH were obtained from the Medical Birth Registry, a national health registry containing information about all births in Norway. Sample quality control and imputation has previously been described[38]. In brief, individuals were genotyped using different Illumina arrays (HumanCoreExome-12 v.1.1, HumanCoreExome-24 v.1.0, Global Screening Array v.1.0, InfiniumOmniExpress-24 v.2, HumanOmniExpress-24 v.1.0). Individual-level quality control was performed to remove ancestry outliers and individuals with sex discrepancy and call rates of <0.98. Furthermore, SNPs with a minor allele frequency (MAF) of <1%, deviating from Hardy–Weinberg equilibrium ($P < 1 \times 10^{-4}$) or a call rate of <0.98 were removed. Imputation was done using SHAPEITv2 + PBWT on the Sanger imputation server, with HRC v.1.1 as the imputation reference panel. Association analysis was done using regenie[39]. All study participants provided a signed informed consent form, and the study protocol was approved by the administrative board of the MoBa, led by the Norwegian Institute of Public Health. The establishment of MoBa and initial data collection was based on a license from the Norwegian Data Protection Agency and approval from The Regional Committee for Medical Research Ethics. The study was approved by the Norwegian Regional Committee for Medical and Health Research Ethics South-East (2015/2425) and by the Swedish Ethical Review Authority (Dnr 2022-03248-01).

**UK Biobank.** The UK Biobank is a prospective cohort of ~500,000 individuals from across the United Kingdom, recruited between the ages of 40 and 69 years. Genotyping was done in two batches, using the Affymetrix chip UK BiLEVE Axiom87 and Affymetrix UK Biobank Axiom array. Imputation was done using a sample of 150,000 whole-genome-sequenced individuals from the UK Biobank[40]. Only individuals with a registered live or stillbirth (identified through the HESIN delivery table) and of European descent were included in the analysis. Association analysis was performed using software developed at deCODE genetics[31]. The UK Biobank resource was used under application no. 56270. All phenotype and genotype data were collected following an informed consent form obtained from all participants. The North West Research Ethics Committee reviewed and approved UK Biobank's scientific protocol and operational procedures (REC reference no. 06/MRE08/65).

## Phenotype definitions

We divided bleeding in pregnancy into three categories and the following sub-phenotypes:

1. Bleeding in early pregnancy (<20 + 0 gestational weeks)
   a. Bleeding in early pregnancy leading to live birth
   b. Bleeding in early pregnancy ending in any outcome (live birth, pregnancy loss, termination of pregnancy, ectopic pregnancy, molar pregnancy, pregnancy of unknown location)
2. Antepartum bleeding (>20th gestational week, before birth)
3. PPH (PPH, hemorrhage following birth)

   a. PPH caused by atony
   b. PPH caused by retained placenta

We categorized each phenotype using hospital admission codes, although not all codes were available in all countries. We provide a phenotype definition list in Supplementary Table 13. We adjusted analyses for age, parity, gestational duration and weight of the child, if possible. Women with known coagulation disorders were excluded (ICD-10 codes D66-D69, O46.0, O67.0). Furthermore, we excluded multifold pregnancies for antepartum bleeding and PPH, if possible. Lastly, we excluded pregnancies delivered by cesarean section in the PPH analysis, if possible. Exclusion of multifold pregnancies and delivery by cesarean section were only applicable in the CHB and MoBa cohorts.

## Meta-analysis

For the meta-analyses, we combined GWASs from the respective cohorts using a fixed-effects inverse variance method based on effect estimates and standard errors in which each dataset was assumed to have a common odds ratio but was allowed to have different population frequencies for alleles and genotypes. Sequence variants were mapped to NCBI Build38 and matched on position and alleles to harmonize the datasets. After excluding variants with discrepant allele frequency between cohorts, variants with MAF < 0.001% in all cohorts or variants only present in one dataset, 18,009,056 variants were included in the meta-analysis. The threshold for genome-wide significance was corrected for multiple testing with a weighted Bonferroni adjustment that controls for the family-wise error rate, using as weights the enrichment of variant classes with predicted functional impact among association signals[41]. The significance threshold then becomes $4.56 \times 10^{-7}$ for high-impact variants (including stop-gained, frameshift, splice acceptor or donor), $9.12 \times 10^{-8}$ for moderate-impact variants (including missense, splice-region variants and in-frame indels), $8.28 \times 10^{-9}$ for low-impact variants (synonymous, 5′ and 3′ untranslated regions, upstream and downstream variants), $4.19 \times 10^{-9}$ for other DNase I hypersensitivity site variants and $1.38 \times 10^{-9}$ for other non-DNase I hypersensitivity variants. In a random-effects method, a likelihood ratio test was performed in all genome-wide associations (GWAs) to test the heterogeneity of the effect estimate in the four datasets; the null hypothesis is that the effects are the same in all datasets and the alternative hypothesis is that the effects differ between datasets.

## Conditional analysis

Conditional association analyses were performed on the GWASs from Iceland, the UK and Denmark using true imputed genotypes of participants. Approximate conditional analyses (COJO), implemented in the software GCTA, were applied to the lead variants in the Finnish, Estonian and MoBa summary statistics[42,43]. LD between variants was estimated using a set of 5,000 whole-genome-sequenced Icelanders. The analyses were restricted to variants within 1 Mb from the index variants. The $P$ values were combined for all six datasets to identify any secondary signals. Based on the number of variants tested, we required secondary signals to pass a threshold of $P < 5 \times 10^{-8}$ after correcting for the lead variant.

## Comparison of effect sizes for retained placenta and uterine atony

We compared effect sizes for retained placenta and uterine atony by doing a case–case analysis of the summary statistics using ReAct[44]. Only genome-wide significant SNPs found in the main analysis of PPH were included. We assumed no overlap between cases and a full overlap between controls.

## Lookup of variants

Variants and variants in strong LD were looked up in the GWAS catalog to identify prior associations to other phenotypes, using the LDlinkR package[10,45]. Furthermore, we investigated the association of the variants to endometriosis and uterine fibroids in the FinnGen cohort (release 10). The analysis was part of the FinnGen core analysis, done using regenie, in which the analysis was adjusted for age, the first ten principal components, genotyping chip and batch[39]. We adjusted $P$ values for the number of phenotypes (2) and variants (234) tested ($P < 0.05 / (2 \times 234) = 0.0001$).

## Mapping of GWA signals to non-coding annotations

We downloaded annotations of cCREs (v.3) from the ENCODE project (screen.encodeproject.org)[10]. We then determined whether the lead PPH sequence variant or any of their correlated variants ($r^2 > 0.80$); that is, PPH signals, were located within cell-type agnostic cCREs, and cCREs defined in tissue samples relevant to PPH; that is, uterus tissue. In this same way, we annotated the PPH signals with respect to enhancer elements (active/genic) as defined for 833 samples (representing 33 groups of tissues and organs) in EpiMap (compbio.mit.edu/epimap)[12]. EpiMap further provides predicted links between enhancers and genes, and, based on these precomputed predictions, we looked for candidate gene targets for each signal in uterus tissue (personal.broadinstitute.org/cboix/epimap/links/links_corr_only). We also annotated the PPH signals with respect to DNA binding sites for 1,210 transcription factors mapped experimentally by various researchers, notably the ENCODE project, using chromatin immunoprecipitation sequencing (ChIP–seq) in different tissue and cell types and conditions made available by Remap2022 (remap2022.univ-amu.fr), which amount to a total of 4,143 ChIP–seq experiments.

## Enrichment of association signals in functional annotations

We used GWA signals from the GWAS catalog (see details in next paragraph) to obtain the null distribution in our enrichment analyses for functional annotations of the genome. The number of sequence variants found in high LD ($r^2 > 0.80$) for each of the five PPH association signals was expected to influence the probability of finding an overlap to a given functional annotation map. We therefore randomly selected five GWA signals from the GWAS catalog for each of the five PPH signals, ensuring that the five randomly selected signals were matched to the PPH signals with respect to the number of sequence variants found in high LD. We then counted the number of randomly selected signals that intersected with a given annotation (this count is denoted as $z$). This procedure was then repeated $n = 200,000$ times. In summary, we were simulating the five PPH signals in terms of the number of sequence variants in high LD to each PPH signal and the property of being a GWA signal associated with a human multifactorial trait.

Let $z_i$ represent the number of annotated signals in each $i$-th sample. The probability ($p$) of finding an intersection to a given annotation among randomly sampled GWA signals is therefore $p = \frac{\sum_i^N z_i}{5N}$, where $5N$ is the total number of randomly sampled GWA signals from the GWAS catalog (five randomly selected GWA signals in each of $N$ samples); this is the expected proportion of annotated GWA signals. We then define $X \sim Bin(n, p)$ where $X$ is the number of annotated PPH signals and $n$ is the number of PPH signals ($n = 5$). The five PPH signals are found on different chromosomes, and we therefore assume that they are

independent. We then determine the probability of observing $x$ or more PPH signals in a given annotation, where $x$ is the observed number of PPH signals that intersect with the given annotation. We are therefore interested in $(X \geq x) = j / N$, where $j$ is the number of times we found $x$ or more annotated GWA signals in the aforementioned $N$ random samples of GWA signals. We then used Bonferroni correction to set the threshold for significance.

For the GWAS catalog, we compiled a robust set of association signals from the NHGRI–EBI catalog of GWAS association signals; downloaded on 4 Aug 2021 (GWAS catalog v.1.00; www.ebi.ac.uk/gwas)[10]. GWAS catalog variants (lead) were matched to in-house variant calls on the basis of rs-identifiers, genome position and MAF (GWAS catalog entries with missing information in any of these fields were omitted). In the GWAS catalog, the same trait has been studied by many different research groups and therefore many associations are 'repeated' and thus not independent. We used the following procedure to compile a set of independent associations for each trait in the GWAS catalog. First, we extracted all associations with the trait with $P < 1 \times 10^{-9}$. Second, we selected the most significant association and added it to the list of independent associations. Third, we added the most significant associations with $P < 1 \times 10^{-9}$ located more than 1 Mb away from other independent associations. We then repeated this third step until no more associations were found with $P < 1 \times 10^{-9}$ that were also located >1 Mb away from those already added to the list of independent associations. We omitted traits classified as 'blood protein measurement' (mostly representing GWAS for serum protein assays) and 16 other traits (for example, heel bone mineral density) with an unusually large number of associations. Furthermore, as our enrichment method takes LD into account (computed in whole-genome sequenced individuals from the Icelandic population), we selected GWAS carried out in individuals of European descent. This resulted in 27,546 GWA association signals for 1,173 diseases or other human traits.

## Functional enrichment and tissue specificity

We used MAGMA to investigate tissue expression specificity[20]. Preprocessed consensus bulk and single-cell RNA sequencing data were downloaded from the Human Protein Atlas[21]. In short, the HPA consensus tissue gene data summarizes expression at the gene level covering 62 tissues and includes data from the Human Protein Atlas, GTEx and FANTOM5. The RNA single-cell consensus dataset covers 51 cell types across 13 tissues from 14 different studies. We used the 1,000 Genomes Phase 3 European data as a reference (downloaded from https://cncr.nl/research/magma/).

## Comorbidity analysis

Comorbidities associated with early bleeding in pregnancy and PPH were identified across three cohorts (Denmark, EstBB and the UK Biobank). The Danish cohort used nationwide data from the Danish National Patient Register and the Danish Medical Birth Register[46,47]. The Danish National Patient Register contains hospital admissions since 1977, and the Danish Medical Birth Register contains births since 1973. We identified all women born after 1957, which ensured a full reproductive history from their 20th birthday and onwards. We analyzed associations between early bleeding in pregnancy, PPH and all other diagnoses (excluding chapters regarding infections, obstetric diagnosis, injuries and contacts with the healthcare system). Similarly, a phenome-wide association scan was performed in the EstBB and UK Biobank. In the UK Biobank, we included only women present in the HESIN delivery tables. Odds ratios were determined using logistic regression, adjusting for year of birth. Data from the three cohorts were meta-analyzed using an inverse variance weighting as implemented on the R package metafor. We controlled for multiple testing by calculating $q$-values and selecting associations with $q < 0.05$.

## Heritability and genetic correlations

SNP heritability was estimated using RHE-mc[48]. We selected genotyped SNPs in the CHB with MAF > 1%, missing in less than 1% of samples, with no deviation from Hardy–Weinberg equilibrium ($P < 10^{-7}$), and we excluded the major histocompatibility complex region, as per author's recommendations. We adjusted the analysis for year of birth, year of birth squared and the first ten principal components.

Genetic correlations were estimated using LD Score Regression[49]. We selected phenotypes based on prior knowledge about risk factors and associations from the comorbidity analysis and availability. In this analysis, we used results for about 1.2 million well-imputed variants, and for LD information we used precomputed LD scores for European populations (downloaded from https://data.broadinstitute.org/alkesgroup/LDSCORE/eur_w_ld_chr.tar.bz2). Genetic correlation of pregnancy bleeding subtypes was calculated between the Danish primary trait and the meta-analysis of the relevant secondary trait, excluding Danes, and vice versa. The results of the two analyses were then meta-analyzed. Genetic correlation of 'Early bleeding, birth' was only done using the Danish data for the primary trait as the sample size for the remaining populations was too small.

## PRSs

PRSs were created using LDPred2 (ref. 50). Autosomal genotype data from 138,669 individuals in the CHB Study on Reproduction was filtered to only include variants present in LDpred2's recommended set of 1,054,330 reference variants. Missing genotype information was imputed to be the affected locus' reference allele. GWAS summary statistics for birth weight from a previous publication[51] were preprocessed with MungeSumStats[52]. The birth weight summary statistics contain a very small fraction of Danish samples from other cohorts. We excluded any Danes from the summary statistics used for the PPH PRS to avoid inflation.

The effects of PRSs were estimated using a logistic regression model, adjusted for maternal age at conception, parity, pre-pregnancy body mass index, previous number of cesarean sections and previous numbers of PPH events. We compared models with and without PRSs using a likelihood ratio test. Furthermore, we also compared the C-index and Nagelkerke's $R^2$. We used a bootstrap resampling approach to find optimism-corrected values, which is a conservative method for internal validation[53]. We repeated the bootstrap resampling 100 times, and we report the 95% percentile bootstrap confidence intervals. We used the Huber–White method to correct standard errors for the inherent clustering present owing to multiple pregnancies from the same women.

## Haplotype analysis

We explored whether the effects of the identified variants on PPH depend on maternal, fetal or maternal and fetal origins by performing an association analysis using the parental transmitted and nontransmitted alleles. We used phased genotype data from the MoBa cohort ($n = 22,330$ parent–offspring trios) and the deCODE study to infer the parent of origin of fetal alleles. The analysis of the deCODE data was done on 106,622 parent–offspring trios (2,558 cases and 104,064 controls) with at least one genotyped individual. This included 19,488 fully genotyped trios, 5,991 with only child and mother and 1,835 with only child and father genotyped, 39,390 with both parents genotyped but not the child, and 1,661, 26,582 and 11,675 with only child, mother or father genotyped, respectively.

For each lead variant, the following logistic regression model was fit: $PPH = MnT + MT + PnT + PT$ + covariates where $MnT$ and $MT$ refer to the maternal nontransmitted and transmitted alleles, respectively, and $PnT$ and $PT$ refer to the paternal nontransmitted and transmitted alleles, respectively. The $PT$ effect is interpreted as a fetal-only genetic effect, whereas the effect of the maternal nontransmitted allele is a maternal-only genetic effect. In the deCODE study, we used maximum

likelihood estimation to estimate the effects, as previously described[54]. Estimates from the two cohorts were meta-analyzed using fixed-effect meta-analysis.

## Statistics and reproducibility
No statistical method was used to predetermine sample size. Individuals of non-European ancestry were excluded in this analysis. No data were excluded from the analyses. The experiments were not randomized. The investigators were not blinded to allocation during experiments and outcome assessment.

## Reporting summary
Further information on research design is available in the Nature Portfolio Reporting Summary linked to this article.

## Data availability
Meta-analysis summary statistics for PPH (including all subtypes), early bleeding (including all subtypes) and antepartum hemorrhage are deposited at https://www.decode.com/summarydata. URLs for other external data used are as follows. Annotations of cCREs, screen.encodeproject.org; EpiMap, compbio.mit.edu/epimap; Remap2022, remap2022.univ-amu.fr; GWAS catalog, https://www.ebi.ac.uk/gwas/; precomputed LD scores for European populations, https://data.broadinstitute.org/alkesgroup/LDSCORE/eur_w_ld_chr.tar.bz2; Human Protein Atlas, https://www.proteinatlas.org/about/download; NCBI Build 38, https://www.ncbi.nlm.nih.gov/. Source data are provided with this paper.

## Code availability
We used publicly available software: GCTA (v.1.91.1 beta, https://yanglab.westlake.edu.cn/software/gcta); ReAct (https://github.com/Paschou-Lab/ReACt, commit 3e285901529628551a078cf99e26a8879714c26d); LDlinkR (v.1.3.0, https://github.com/CBIIT/LDlinkR); MAGMA (v.1.10, https://ctg.cncr.nl/software/magma); RHE-mc (https://github.com/sriramlab/RHE-mc, commit a3dc6eab08ede92e711ed5532e8aad4708225c95); plink (v.1.9, https://www.cog-genomics.org/plink); LD Score Regression (https://github.com/bulik/ldsc, commit aa33296abac9569a6422ee6ba7eb4b902422cc74); LDPred2 (v.1.10, https://github.com/privefl/bigsnpr); ggplot2 (v.3.3.3, https://ggplot2.tidyverse.org); and R (v.4.3, https://www.r-project.org). No custom code or software was used for any aspect of the study.

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

## Acknowledgements
The work is carried out as a part of the BRIDGE—Translational Excellence Programme (bridge.ku.dk) at the Faculty of Health and Medical Sciences, University of Copenhagen, funded by the Novo Nordisk Foundation under grant agreements NNF18SA0034956, NNF14CC0001, NNF22OC0077221 and NNF17OC0027594. We would also like to acknowledge funding from the Ole Kirk Foundation, A. P. Moller Foundation, and Rigshospitalet's Research Fund. B.J. received funding from The Swedish Research Council, Stockholm, Sweden (2019-01004), The Research Council of Norway, Oslo,

Norway (FRIMEDBIO no. 547711), March of Dimes (no. 21-FY16-121) and Agreement concerning research and education of doctors (ALFGBG-965353). Research by B.J. was also supported by the Eunice Kennedy Shriver National Institute Of Child Health & Human Development of the National Institutes of Health under award number R01HD101669. The content is solely the responsibility of the authors and does not necessarily represent the official views of the National Institutes of Health. We thank the Norwegian Institute of Public Health for generating high-quality genomic data. This research is part of the HARVEST collaboration, supported by the Research Council of Norway (no. 229624). We also thank deCODE genetics and the NORMENT Centre for providing genotype data, funded by the Research Council of Norway (no. 223273), South East Norway Health Authorities and Stiftelsen Kristian Gerhard Jebsen. We further thank the Center for Diabetes Research, the University of Bergen for providing genotype data and performing quality control and imputation of the data funded by the European Research Council Advanced Grant project SELECTionPREDISPOSED, Stiftelsen Kristian Gerhard Jebsen, Trond Mohn Foundation, the Research Council of Norway, the Novo Nordisk Foundation, the University of Bergen and the Western Norway Health Authorities. We acknowledge the participants and investigators of the FinnGen study. The FinnGen project is funded by two grants from Business Finland (HUS 4685/31/2016 and UH 4386/31/2016) and the following industry partners: AbbVie, AstraZeneca UK, Biogen MA, Bristol Myers Squibb (and Celgene Corporation and Celgene International II Sàrl), Genentech, Merck Sharp & Dohme, Pfizer, GlaxoSmithKline Intellectual Property Development, Sanofi US Services, Maze Therapeutics, Janssen Biotech, Novartis and Boehringer Ingelheim International. The following biobanks are acknowledged for delivering biobank samples to FinnGen: Auria Biobank (www.auria.fi/biopankki), THL Biobank (www.thl.fi/biobank), Helsinki Biobank (www.helsinginbiopankki.fi), Biobank Borealis of Northern Finland (https://www.ppshp.fi/Tutkimus-ja-opetus/Biopankki/Pages/Biobank-Borealis-briefly-in-English.aspx), Finnish Clinical Biobank Tampere (www.tays.fi/en-US/Research_and_development/Finnish_Clinical_Biobank_Tampere), Biobank of Eastern Finland (www.ita-suomenbiopankki.fi/en), Central Finland Biobank (www.ksshp.fi/fi-FI/Potilaalle/Biopankki), Finnish Red Cross Blood Service Biobank (www.veripalvelu.fi/verenluovutus/biopankkitoiminta), Terveystalo Biobank (www.terveystalo.com/fi/Yritystietoa/Terveystalo-Biopankki/Biopankki/) and Arctic Biobank (https://www.oulu.fi/en/university/faculties-and-units/faculty-medicine/northern-finland-birth-cohorts-and-arctic-biobank). All Finnish biobanks are members of the Biobanking and BioMolecular Resources Infrastructure (www.bbmri.fi); the Finnish Biobank Cooperative (https://finbb.fi) is the coordinator of BBMRI-ERIC operations in Finland. The Finnish biobank data can be accessed through the Fingenious services (https://site.fingenious.fi/en) managed by the Finnish Biobank Cooperative. This Estonian Biobank study was funded by the European Union through the European Regional Development Fund project no. 2014-2020.4.01.15-0012 GENTRANSMED. Data analysis was carried out in part in the High-Performance Computing Center of the University of Tartu. We acknowledge the Estonian Biobank research team, including A. Metspalu, L. Milani, R. Mägi, M. Nelis and G. Hudjashov, for data collection, genotyping, quality control and imputation.

## Author contributions

D.W., S.B. and H.S.N. conceived the study. S.R.O., O.B.P., C.E., E.S., C.M., M.T.B., B.A.J., T.B., H.U., A.S.H., P.M., O.A.A., P.R.N., B.J., A.M.K., L.K., X.W., F.G., B.F., P.D.R., M.N., T.F.H., M.D., C.L., K.B., U.T. and H.L. carried out data collection, subject ascertainment, recruitment and/or phenotyping. D.W., V.S., L.S. and G.T. performed the GWAS meta-analyses. D.W., V.S., L.S., T.L., C.F., X.W., O.S., G.S. and D.G. performed other analysis. D.W., V.S., S.B. and H.S.N. wrote the paper with input from all authors. K.S., S.B. and H.S.N. supervised the study. All authors approved the final version of the paper.

## Competing interests

H.S.N. obtained speaker fees from Ferring Pharmaceuticals, Merck, AstraZeneca, Novo Nordisk and Cook Medical. S.B. has obtained speaker fees from MDS, Bayer, LEO Pharma and Astellas and has ownership in Hoba Therapeutics, Novo Nordisk, Lundbeck, ALK, Eli Lilly and managing board memberships in Proscion and Intomics. All authors affiliated with deCODE genetics are employees of deCODE genetics, a subsidiary of Amgen. The other authors declare no competing interests.

## Additional information

**Correspondence and requests for materials** should be addressed to Søren Brunak or Henriette Svarre Nielsen.

# Reporting Summary

## Statistics

For all statistical analyses, confirm that the following items are present in the figure legend, table legend, main text, or Methods section.

| n/a | Confirmed | |
|---|---|---|
| ☐ | ☒ | The exact sample size (*n*) for each experimental group/condition, given as a discrete number and unit of measurement |
| ☐ | ☒ | A statement on whether measurements were taken from distinct samples or whether the same sample was measured repeatedly |
| ☐ | ☒ | The statistical test(s) used AND whether they are one- or two-sided<br>*Only common tests should be described solely by name; describe more complex techniques in the Methods section.* |
| ☐ | ☒ | A description of all covariates tested |
| ☐ | ☒ | A description of any assumptions or corrections, such as tests of normality and adjustment for multiple comparisons |
| ☐ | ☒ | A full description of the statistical parameters including central tendency (e.g. means) or other basic estimates (e.g. regression coefficient) AND variation (e.g. standard deviation) or associated estimates of uncertainty (e.g. confidence intervals) |
| ☐ | ☒ | For null hypothesis testing, the test statistic (e.g. *F*, *t*, *r*) with confidence intervals, effect sizes, degrees of freedom and *P* value noted<br>*Give P values as exact values whenever suitable.* |
| ☒ | ☐ | For Bayesian analysis, information on the choice of priors and Markov chain Monte Carlo settings |
| ☒ | ☐ | For hierarchical and complex designs, identification of the appropriate level for tests and full reporting of outcomes |
| ☐ | ☒ | Estimates of effect sizes (e.g. Cohen's *d*, Pearson's *r*), indicating how they were calculated |

*Our web collection on statistics for biologists contains articles on many of the points above.*

## Software and code

Policy information about availability of computer code

| Data collection | No software was used for data collection |
|---|---|
| Data analysis | We used publicly available software:<br>GCTA (v1.91.1 beta, https://yanglab.westlake.edu.cn/software/gcta/)<br>ReAct (https://github.com/Paschou-Lab/ReACt, commit 3e285901529628551a078cf99e26a8879714c26d)<br>LDlinkR (v1.3.0, https://github.com/CBIIT/LDlinkR)<br>MAGMA (v1.10, https://ctg.cncr.nl/software/magma)<br>RHE-mc (https://github.com/sriramlab/RHE-mc, commit a3dc6eab08ede92e711ed5532e8aad4708225c95)<br>plink (v1.9, https://www.cog-genomics.org/plink/)<br>LDSCore Regression (https://github.com/bulik/ldsc, commit aa33296abac9569a6422ee6ba7eb4b902422cc74<br>LDPred2 (v1.10, https://github.com/privefl/bigsnpr)<br>ggplot2 (v3.3.3, https://ggplot2.tidyverse.org/)<br>R (v4.3, https://www.r-project.org) |

For manuscripts utilizing custom algorithms or software that are central to the research but not yet described in published literature, software must be made available to editors and reviewers. We strongly encourage code deposition in a community repository (e.g. GitHub). See the Nature Portfolio guidelines for submitting code & software for further information.

# Data

Policy information about availability of data

All manuscripts must include a data availability statement. This statement should provide the following information, where applicable:

- Accession codes, unique identifiers, or web links for publicly available datasets
- A description of any restrictions on data availability
- For clinical datasets or third party data, please ensure that the statement adheres to our policy

Meta-analysis summary statistics are deposited at https://www. decode.com/summarydata/. FinnGen data are publicly available and were downloaded from https://www.finngen.fi/en/access_results.

URLs for other external data used are as follows: Annotations of candidate cis-regulatory elements, screen.encodeproject.org; EpiMap, compbio.mit.edu/epimap; Remap2022, remap2022.univ-amu.fr; GWAS Catalog, https://www.ebi.ac.uk/gwas/; precomputed LD scores for European populations, https://data.broadinstitute.org/alkesgroup/LDSCORE/eur_w_ld_chr.tar.bz2; Human Protein Atlas, https://www.proteinatlas.org/about/download; NCBI Build 38, https://www.ncbi.nlm.nih.gov/.

# Research involving human participants, their data, or biological material

Policy information about studies with human participants or human data. See also policy information about sex, gender (identity/presentation), and sexual orientation and race, ethnicity and racism.

| Reporting on sex and gender | The study has focused on the maternal genetics of bleeding in pregnancy, and it must be expected that the mains findings only apply to women. However, we have also used paternal data in the analysis of fetal and maternal transmission. |
| --- | --- |
| Reporting on race, ethnicity, or other socially relevant groupings | *Please specify the socially constructed or socially relevant categorization variable(s) used in your manuscript and explain why they were used. Please note that such variables should not be used as proxies for other socially constructed/relevant variables (for example, race or ethnicity should not be used as a proxy for socioeconomic status).* *Provide clear definitions of the relevant terms used, how they were provided (by the participants/respondents, the researchers, or third parties), and the method(s) used to classify people into the different categories (e.g. self-report, census or administrative data, social media data, etc.)* *Please provide details about how you controlled for confounding variables in your analyses.* |
| Population characteristics | All women with at least one pregnancy from each of the participating biobanks were included. A description of the population characteristics is included in the Methods section. |
| Recruitment | The UK Biobank project is a large-scale prospective cohort study that includes approximately 500,000 individuals from various regions of the United Kingdom . The Icelandic deCODE Genetics study consists of participants recruited through multiple research projects conducted at deCODE Genetics. Finngen is a biobank that incorporates both legacy samples, initially collected by the National Institute for Health and Welfare in Finland, and prospective samples acquired from hospital biobanks. The Copenhagen Hospital Biobank Reproduction Study (CHB-REPRO) is a specialized sub-cohort that focuses on patients with reproductive disorders, drawn from the Copenhagen Hospital Biobank, which itself is based on patient blood samples collected in Danish hospitals. The Estonian Biobank is a population-based collection that boasts over 200,000 participants, amounting to roughly 20% of the entire Estonian population. The Norwegian Mother and Child Cohort Study (MoBa) is a population-based pregnancy cohort study executed by the Norwegian Institute of Public Health. The recruitment phase spanned from 1999 to 2008 and included participants from diverse regions of Norway. |
| Ethics oversight | The deCODE study was approved by the Icelandic National Bioethics Committee (VSN-15-169). The North West Research Ethics Committee reviewed and approved UK Biobank's scientific protocol and operational procedures (REC reference no.: 06/MRE08/65). Approval of the Copenhagen Hospital Biobank Reproductive Health Study (CHBRHS) was obtained from the Danish National Committee on Health Research Ethics (NVK-1805807) and the Capital Region Data Protection Agency (P-2019-49). All study participants provided a signed informed consent, and the study protocol has been approved by the administrative board of the Norwegian Mother, Father and Child Cohort Study, led by the Norwegian Institute of Public Health. The establishment of MoBa and initial data collection was based on a license from the Norwegian Data Protection Agency and approval from The Regional Committee for Medical Research Ethics. The study was approved by the Norwegian Regional Committee for Medical and Health Research Ethics South-East (2015/2425) and by the Swedish Ethical Review Authority (Dnr 2022-03248-01). Participants in FinnGen provided informed consent for biobank research on basis of the Finnish Biobank Act. Alternatively, separate research cohorts, collected before the Finnish Biobank Act came into effect (in September 2013) and the start of FinnGen (August 2017) were collected on the basis of study-specific consent and later transferred to the Finnish biobanks after approval by Fimea, the National Supervisory |

Authority for Welfare and Health. Recruitment protocols followed the biobank
protocols approved by Fimea. The Coordinating Ethics Committee of the Hospital
District of Helsinki and Uusimaa (HUS) approved the FinnGen study protocol
(number HUS/990/2017). The FinnGen study is approved by the Finnish Institute for
Health and welfare (approval numberTHL/2031/6.02.00/2017, amendments
TH L/1101/5.05.00/2017, THL/341/6.02.00/2018, THL/2222/6.02 .00/2018,
THL/283/6.02 .00/2019 and THL/1721/5.05.00/2019), the Digital and Population Data
Service Agency (VRK43431/2017-3, VRK/6909/2018-3 and VRK/4415/2019-3), the
Social Insurance Institution (KELA) (KELA 58/522/2017, KELA 131/522/2018,
KELA 70/522/2019 and KELA 98/522/2019) and Statistics Finland (TK-53-1041-17).
The activities of the Est BB are regulated by the Human Genes Research Act, which
was adopted in 2000 specifically for the operations of the Est BB. All Estonian
Biobank participants have signed a broad informed consent form and analyses were
carried out under ethical approval 1.1-12/624 from the Estonian Committee on Bioethics and Human Research (Estonian
Ministry of Social Affairs) and data release
NOS from the EstBB.

Note that full information on the approval of the study protocol must also be provided in the manuscript.

# Field-specific reporting

Please select the one below that is the best fit for your research. If you are not sure, read the appropriate sections before making your selection.

☒ Life sciences ☐ Behavioural & social sciences ☐ Ecological, evolutionary & environmental sciences

For a reference copy of the document with all sections, see nature.com/documents/nr-reporting-summary-flat.pdf

# Life sciences study design

All studies must disclose on these points even when the disclosure is negative.

| | |
|---|---|
| Sample size | GWAS meta analysis. We combined available data from all cohorts. |
| Data exclusions | The study included all available data except for participants from non-European ethnicities, consistent with the approach taken for all groups. |
| Replication | The analysis reported here consist of all data from the six different populations, and replication was not performed as the analysis included all data. |
| Randomization | No randomization was performed. Relevant covariates were included in GWA analysis to account for potential confounding. |
| Blinding | Group allocation was not relevant to this study, hence blinding was not performed. |

# Reporting for specific materials, systems and methods

We require information from authors about some types of materials, experimental systems and methods used in many studies. Here, indicate whether each material, system or method listed is relevant to your study. If you are not sure if a list item applies to your research, read the appropriate section before selecting a response.

## Materials & experimental systems

| n/a | Involved in the study |
|---|---|
| ☒ | ☐ Antibodies |
| ☒ | ☐ Eukaryotic cell lines |
| ☒ | ☐ Palaeontology and archaeology |
| ☒ | ☐ Animals and other organisms |
| ☒ | ☐ Clinical data |
| ☒ | ☐ Dual use research of concern |
| ☒ | ☐ Plants |

## Methods

| n/a | Involved in the study |
|---|---|
| ☒ | ☐ ChIP-seq |
| ☒ | ☐ Flow cytometry |
| ☒ | ☐ MRI-based neuroimaging |

## Plants

| | |
|---|---|
| Seed stocks | *Report on the source of all seed stocks or other plant material used. If applicable, state the seed stock centre and catalogue number. If plant specimens were collected from the field, describe the collection location, date and sampling procedures.* |
| Novel plant genotypes | *Describe the methods by which all novel plant genotypes were produced. This includes those generated by transgenic approaches, gene editing, chemical/radiation-based mutagenesis and hybridization. For transgenic lines, describe the transformation method, the number of independent lines analyzed and the generation upon which experiments were performed. For gene-edited lines, describe the editor used, the endogenous sequence targeted for editing, the targeting guide RNA sequence (if applicable) and how the editor was applied.* |
| Authentication | *Describe any authentication procedures for each seed stock used or novel genotype generated. Describe any experiments used to assess the effect of a mutation and, where applicable, how potential secondary effects (e.g. second site T-DNA insertions, mosiacism, off-target gene editing) were examined.* |

