## [Peer Review File · Nature Genetics]

Peer Review Information

Manuscript Title: Genome-wide association meta-analysis identifies five loci associated with postpartum hemorrhage

Corresponding author name(s): Professor Henriette (Svarre) Nielsen, Professor Søren Brunak

Reviewer Comments & Decisions:

Decision Letter, initial version:

12th Oct 2023

Dear Professor Nielsen,

First, please accept my apologies for the delay in returning this decision to you. Thank you for bearing with me.

Your Article, "Pregnancy-Associated Bleeding and Genetics: Five Sequence Variants in the Myometrium and Progesterone Signaling Pathway are associated with postpartum hemorrhage" has now been seen by 3 referees. You will see from their comments below that while they find your work of interest, some important points are raised. We are interested in the possibility of publishing your study in Nature Genetics, but would like to consider your response to these concerns in the form of a revised manuscript before we make a final decision on publication.

To guide the scope of the revisions, the editors discuss the referee reports in detail within the team, including with the chief editor, with a view to identifying key priorities that should be addressed in revision and sometimes overruling referee requests that are deemed beyond the scope of the current study. We hope that you will find the prioritized set of referee points to be useful when revising your study. Please do not hesitate to get in touch if you would like to discuss these issues further.

We therefore invite you to revise your manuscript taking into account all reviewer and editor comments. Please highlight all changes in the manuscript text file. At this stage we will need you to upload a copy of the manuscript in MS Word .docx or similar editable format.

*2) If you have not done so already please begin to revise your manuscript so that it conforms to our Article format instructions, available here.
Refer also to any guidelines provided in this letter.

Please be aware of our guidelines on digital image standards.

[redacted]

We hope to receive your revised manuscript within four to eight weeks. If you cannot send it within this time, please let us know.

Sincerely,

Safia Danovi
Editor
Nature Genetics

Referee expertise:

Referee #1: genetics, pregnancy

Referee #2: GWAS, reproductive health

Referee #3: GWAS, fetal/maternal health

Reviewers' Comments:

Reviewer #1:

Remarks to the Author:

This manuscript describes a GWAS of bleeding during pregnancy and postpartum haemorrhage, which have strong implications for fetal and maternal health, respectively. The study makes an important contribution to knowledge of the genetics of these conditions. Data from 6 northern-European cohorts was assembled, and rigorous statistical methods were employed. A suite of informative follow up analyses showed that the 5 signals identified for PPH implicate hormone regulation in its etiology and show little evidence of heterogeneity between PPH due to the known causes of uterine atony vs. retained placenta. There are clear correlations between the genetics of PPH and the genetics of uterine fibroids and endometriosis. On the other hand, the genetics of early pregnancy bleeding was not well correlated with that of PPH and the extensive number of genetic correlations with other traits suggest a much more heterogeneous phenotype.

I have the following, generally minor comments.

1. As a non-clinician reader, I would appreciate a little more help in the main text to understand the phenotypes. Fig 1 is helpful, but does not clarify everything. Fig. 1 lists "early pregnancy bleeding", "anteartum bleeding" and "post-partum bleeding", but the GWAS text refers to "anteartum haemorrhage" and "postpartum haemorrhage" as the three main phenotypes examined. Presumably haemorrhage has a more specific/potentially more severe meaning than "bleeding" referred to in fig. 1. Can the authors clarify? Is there a helpful definition of haemorrhage, which separates this from any bleeding? – is there a threshold of bleeding below which an individual would not be included as a case in the early, anteartum or postpartum analyses? I appreciate that the methods section and supplementary table 13 indicates phenotype definitions and clinical codes, but it would be helpful to include a clear definition of "haemorrhage" and "bleeding" phenotypes in the manuscript introduction. (NB where the phenotype description methods refer to sup table 12, I think this should be sup table 13.)
2. Related to the above, the article should be checked for instances where "postpartum bleeding" is used in place of "postpartum haemorrhage" (e.g. legend of fig 3), and corrected for precision.
3. Was there overlap between the cases? Were some women included in both the early bleeding and PPH case groups, for example? Did the controls all have no bleeding at all in pregnancy or post-pregnancy? Depending on the extent of overlap of cases and controls, a Venn diagram would be

helpful to enable the reader to gauge how independent these phenotype groups are.

4. Fig 1B: are the labels showing the names of the nearest genes to the top SNP? – it would be helpful to clarify what these are in the legend, and also add similar labels to Fig. 1C so that the reader can see whether they are the same or different signals.

5. Sup fig 1: the cohort names should be written in full in the legend so that the reader can understand what the abbreviations mean.

6. Overall findings: "In addition, we analyzed uterine atony (13,048 cases and 261,809 controls) and retained placental tissue (6,256 cases and 266,427 controls), where three (chromosome 4, 6, and 10) and one (chromosome X) of the five associated loci passed multiple testing correction, respectively (Figure 1C, Supplementary Table 2)." Here, it would be helpful to clarify that this was a sub-analysis of the main PPH analysis. Perhaps clarify as "we analysed PPH as a result of uterine atony and PPH as a result of retained placental tissue".

7. Where Figure 2B is mentioned in the overall findings section, I think the authors meant to say "Figure 1B".

8. Figure 2: it would be helpful to clarify that this is about the results of the PPH GWAS analysis (not the other phenotypes).

9. Methods: heritability and genetic correlations, paragraph 1: should "HMC" be "MHC"?

10. It would be helpful to add a column for "Other allele" to table 1.

11. What do the error bars represent on supplementary figure 5?

12. Sup table 1: it is interesting that for most cohorts, the numbers for PPH – retained placenta are lower than those for PPH – uterine atony, but for FINNGEN, they are higher. Can the authors comment on why this might be?

13. The genetic correlations with birth weight and gestational duration are interesting. Can the authors comment on possible mechanisms underlying these correlations? For example, are longer pregnancies and larger babies likely to contribute causally to a higher risk of PPH?

Reviewer #2:

Remarks to the Author:

This is an important study that analysed genetic risk factors contributing to variation in pregnancy-associated bleeding. Post-partum hemorrhage (PPH) is a leading cause of death following childbirth although pregnancy associated bleeding can occur at all stages of pregnancy. PPH results in ~100,000 deaths of young and otherwise healthy women each year. Early identification of women at risk for PPH together with appropriate management could reduce the mortality and morbidity associated with PPH. No associations were observed for bleeding in early pregnancy or antepartum hemorrhage. The study identified five genome-wide significant signals associated with PPH. Functional annotation identified HAND2, TBX3, and RAP2C/FRMD7 as likely candidate genes at three loci and showed that associated

variants at each locus were located within binding sites for progesterone receptors.

The methods provide general phenotypic definitions for bleeding during pregnancy and PPH sub-phenotype categories defined using hospital admission codes with details provided in Supplementary Table 13. However, the authors also note that not all codes were available in all countries. Detailed descriptions on what information was recorded, how participants were ascertained and inclusion/exclusion criteria for the individual studies should be provided. What information was available to exclude pregnancies delivered by caesarean section for the PPH analyses and did this vary across the cohorts? How accurate was the ascertainment of cases with retained placenta or uterine atony and did this vary across cohorts? Would variation in ascertainment for the different phenotypes across the different cohorts be likely to influence the results reported?

The authors report enrichment of PPH signals in PGR binding sites in human embryonic stem cells and nominal significance in the myometrium. The role of dysregulation in the myometrium is highlighted in the title and abstract. What other evidence is there for a role of these genes in regulation in myometrium affecting PPH?

What is known about the role of progesterone and PGR in the regulation of reported candidate genes? Progesterone is important in the establishment and maintenance of pregnancy. Could factors during the establishment of pregnancy influence the frequency of PPH or is the role of these genes restricted to effects around parturition?

The authors report that some PPH variants were also associated with endometriosis and/or uterine fibroids and an inverse genetic correlation between PPH and uterine fibroids. What is the significance of the highlighted genes and reported overlaps between PPH and endometriosis, and the inverse correlation uterine fibroids?

The PRS provided a marginal improvement in prediction for PPH. Is family history a risk factor? What increase in power for genetic studies will be required before PRS prediction for PPH is clinically relevant?

In the second paragraph of the results referring to the associations with endometriosis and/or uterine fibroids (the text refers readers to Table 2, but this should be Table 1).

The legend to Table 1 states that endometriosis and uterine fibroids estimates come from the datasets listed in Supplementary Table 4. However, Supplementary Table 4 just has a general reference to the GWAS Catalog. What datasets were used for the endometriosis and uterine fibroids estimates in Table 1?

Reviewer #3:

Remarks to the Author:

Perinatal bleeding, including early gestation and postpartum hemorrhage (PPH), is an important public health problem associated with maternal death and many co-morbidities. In a genome-wide association study meta-analysis of cohorts involving European ancestry individuals, Westergaard et al found five loci associated with PPH, and demonstrated that the effect of the loci is more likely to be via the maternal rather than the fetal genome. No associations were detected for early pregnancy

bleeding. Ancillary work showed presence of genetic correlations of PPH and early gestational bleeding with other disorders and traits. The prominent discovery is the observation from annotations of the associated nearby genes and downstream analysis pointing to PGR-responsiveness pathways in PPH, partly supporting previously known signals while providing the importance of maternal genetic variation in these pathways and bleeding complications. Comments/suggestions are listed below.

1. The genes linked to PPH support a known role of progesterone/hormones in pregnancy maintenance and bleeding complications, and provide new evidence for role of maternal genetic variation. The added value of the work would be strengthened with further functional experiment in relevant models (e.g., cell lines in relevant tissues for a more mechanistic understanding of the tissue/cell type, and regulatory elements impacted). This can shed light on mechanisms of the genetic effect, given previous work establishing the PGR-receptor signals in maternal-fetal hormonal/immunological crosstalk and related pathologies.
2. The heritability estimates do not dissociate maternal and fetal contributions. How much do maternal and fetal genotypes contribute to heritability?
3. How much do the maternal genetic effects of the five loci identified explain the variation in PPH? Is the variance explained strong enough to consider PRS of PPH in prediction?
4. The PRS finding lumps genetic effects on birthweight with PPH, making it difficult to know whether the improvement in prediction is due to PRS of birthweight, PRS of PPH, or both. More importantly, what was the improvement in prediction of PPH when only PRS of PPH (but not PRS of birthweight) was included? Was the PRS of birthweight maternal or fetal? Why was PRS of birthweight selected but not PRS of other traits (e.g. gestational duration, fibroids which are significantly genetically correlated with PPH)?
5. In the Abstract, the last sentence ("...whereas early bleeding is a complex trait related to underlying health and possibly socioeconomic status.") looks an overinterpretation and less substantiated by data, given the study's heritability estimate of 12.7% for early bleeding, which is just a little smaller than the heritability estimate for PPH and shows the impact of genetic variation on the pathology. The observation of genetic correlations with SES factors as presented in Line 241-242 is not adequate to justify this conclusion.
6. In the discussion of lack of signals for early bleeding, phenotypic heterogeneity and polygenicity have been mentioned as potential explanation, but which results/evidence substantiate this explanation is unclear.

Author Rebuttal to Initial comments

Reviewers' Comments:

Reviewer #1:

Remarks to the Author:

This manuscript describes a GWAS of bleeding during pregnancy and postpartum haemorrhage, which have strong implications for fetal and maternal health, respectively. The study makes an important contribution to knowledge of the genetics of these conditions. Data from 6 northern-European cohorts was assembled, and rigorous statistical methods were employed. A suite of informative follow up analyses showed that the 5 signals identified for PPH implicate hormone

regulation in its etiology and show little evidence of heterogeneity between PPH due to the known causes of uterine atony vs. retained placenta. There are clear correlations between the genetics of PPH and the genetics of uterine fibroids and endometriosis. On the other hand, the genetics of early pregnancy bleeding was not well correlated with that of PPH and the extensive number of genetic correlations with other traits suggest a much more heterogeneous phenotype.

I have the following, generally minor comments.

1. As a non-clinician reader, I would appreciate a little more help in the main text to understand the phenotypes. Fig 1 is helpful, but does not clarify everything. Fig. 1 lists “early pregnancy bleeding”, “antepartum bleeding” and “post-partum bleeding”, but the GWAS text refers to “antepartum haemorrhage” and “postpartum haemorrhage” as the three main phenotypes examined. Presumably haemorrhage has a more specific/potentially more severe meaning than “bleeding” referred to in fig. 1. Can the authors clarify? Is there a helpful definition of haemorrhage, which separates this from any bleeding? – is there a threshold of bleeding below which an individual would not be included as a case in the early, antepartum or postpartum analyses? I appreciate that the methods section and supplementary table 13 indicates phenotype definitions and clinical codes, but it would be helpful to include a clear definition of “haemorrhage” and “bleeding” phenotypes in the manuscript introduction. (NB where the phenotype description methods refer to sup table 12, I think this should be sup table 13.)

Answer: The reviewer highlights a discussion we also had internally between clinicians and non-clinicians. Hemorrhage is a more specific term that typically refers to a more substantial or profuse bleeding episode during or postpartum, albeit there is no clear cutoff. For postpartum hemorrhage there are specific definitions used as described in Methods “Phenotype definitions” and in Supplementary Table 13. There are no specific cutoffs for early bleeding or antepartum bleeding. We have clarified this in Figure 1. We have corrected the reference to Supplementary table 13. (Anecdotally, in the Scandinavian languages, it is all just called bleeding).

2. Related to the above, the article should be checked for instances where “postpartum bleeding” is used in place of “postpartum haemorrhage” (e.g. legend of fig 3), and corrected for precision.

Answer: We have corrected the wording systematically, such that it is now the same everywhere, to avoid ambiguity.

3. Was there overlap between the cases? Were some women included in both the early bleeding and PPH case groups, for example? Did the controls all have no bleeding at all in pregnancy or post-pregnancy? Depending on the extent of overlap of cases and controls, a Venn diagram would be helpful to enable the reader to gauge how independent these phenotype groups are.

Answer: There was definitely an overlap between the case groups for early bleeding and postpartum hemorrhage. Early bleeding is a risk factor for postpartum hemorrhage. This has previously been described in the literature (e.g. ref 2). We have included a Venn Diagram describing showing this across the different cohorts (Supplementary Figure 1), and the following text:

“Instances of overlap were observed, and the extent of such overlap exhibited variation across cohorts (Error! Reference source not found.).”

4. Fig 1B: are the labels showing the names of the nearest genes to the top SNP? – it would be helpful to clarify what these are in the legend, and also add similar labels to Fig. 1C so that the reader can see whether they are the same or different signals.

Answer: The labels in Figure 1B show the gene indicated by the functional analysis. We have clarified this in the legends of Figure 1B and Figure 1C. We have also removed the labels from Figure 1C, as the text was too small to read.

5. Sup fig 1: the cohort names should be written in full in the legend so that the reader can understand what the abbreviations mean.

Answer: We have made this correction, thank you. The legend now reads:

“Supplementary Figure 1. (A) Effect sizes in each cohort for the postpartum hemorrhage lead variants, which were largely similar. UK, UK Biobank (England); NO, Norwegian Mother, Father and Child Cohort Study (Norway); IS, deCODE genetics (Iceland); FI, FinnGen (Finland); ES,

Estonian Biobank (Estonia); DK2, Copenhagen Hospital Biobank, years 2012-2018 (Denmark), DK1, Copenhagen Hospital Biobank, years 1977-2011 (Denmark). (B) Genome-wide significant variants from the postpartum hemorrhage analysis are also associated with endometriosis and uterine fibroids. The red line indicates the Bonferroni corrected p-value threshold ($p < 0.05/(2 \text{ loci} * 234 \text{ variants}) = 0.0001$)."

6. Overall findings: "In addition, we analyzed uterine atony (13,048 cases and 261,809 controls) and retained placental tissue (6,256 cases and 266,427 controls), where three (chromosome 4, 6, and 10) and one (chromosome X) of the five associated loci passed multiple testing correction, respectively (Figure 1C, Supplementary Table 2)." Here, it would be helpful to clarify that this was a sub-analysis of the main PPH analysis. Perhaps clarify as "we analysed PPH as a result of uterine atony and PPH as a result of retained placental tissue".

Answer: We have added the suggested text to clarify this:

"In addition, we analyzed PPH as a result of uterine atony (13,048 cases and 261,809 controls) and PPH as a result of retained placental tissue (6,256 cases and 266,427 controls), where three (chromosome 4, 6, and 10) and one (chromosome X) of the five associated loci passed multiple testing correction, respectively (**Error! Reference source not found.C, Error! Reference source not found.**)."

7. Where Figure 2B is mentioned in the overall findings section, I think the authors meant to say "Figure 1B".

Answer: The reviewer is correct; we have changed this.

8. Figure 2: it would be helpful to clarify that this is about the results of the PPH GWAS analysis (not the other phenotypes).

Answer: We have updated the legend to clarify this. The legend now reads:

"Tissue-specific enrichment analysis for postpartum hemorrhage. (A) MAGMA single cell enrichment from the Human Protein Atlas. Smooth muscle cells and endothelial cells were both

enriched (FDR < 0.05) (C) MAGMA bulk tissue enrichment from the Human Protein Atlas showed an enrichment of endometrial, smooth muscle, seminal vesicle, and thyroid gland tissue (FDR < 0.05)."

9. Methods: heritability and genetic correlations, paragraph 1: should "HMC" be "MHC"?

Answer: Yes, it should be MHC, short for major histocompatibility complex. Autocorrect must have played a trick on us. This has been corrected, and it is now spelled out in full:

"... and excluded the major histocompatibility complex region ..."

10. It would be helpful to add a column for "Other allele" to table 1.

Answer: We have made this correction, thank you.

11. What do the error bars represent on supplementary figure 5?

Answer: The error bars represent the 95% confidence interval, calculated from the Wald statistics. We have added this to the legend, and it now reads (NB: Suppl Fig 5 is now Suppl Fig 6 due to addition of the Venn diagram):

"Supplementary Figure 2. Haplotype analysis of the five PPH associated variants in the MoBa and deCODE cohorts. Results suggest that that effect is mediated through the maternal genome. Error bars represent the 95% confidence interval calculated from the Wald statistic. Mnt: maternal non-transmitted; MT: maternal transmitted; PnT: paternal non-transmitted; PT: paternal transmitted"

12. Sup table 1: it is interesting that for most cohorts, the numbers for PPH – retained placenta are lower than those for PPH – uterine atony, but for FINNGEN, they are higher. Can the authors comment on why this might be?

Answer: We are not aware of any biological differences explaining this. It is worth noting that FinnGen is not a population representative sample. It is a mixture of biobanks, combined with recruitment of new individuals. This makes it difficult to compare prevalence across cohorts. We note, however, that the odds-ratio estimated in the GWAS is not affected by this.

13. The genetic correlations with birth weight and gestational duration are interesting. Can the authors comment on possible mechanisms underlying these correlations? For example, are longer pregnancies and larger babies likely to contribute causally to a higher risk of PPH?

Answer: Fetal macrosomia and multiple gestations are believed to dilate the myometrial muscles of the uterus, making it more difficult to contract. The correlation between gestational duration and postpartum hemorrhage is more complex, as it can be both due to biological issues (uterine overdistension and placental issues), or medically induced labour. We have added this to the Discussion:

“Fetal macrosomia and multiple gestations are believed to dilate the myometrial muscles of the uterus, making it more difficult to contract²⁵. The correlation between gestational duration and postpartum hemorrhage is more complex, as it can be both due to biological issues (uterine overdistension and placental issues), or medically induced labor²⁶.”

Reviewer #2:

Remarks to the Author:

This is an important study that analysed genetic risk factors contributing to variation in pregnancy-associated bleeding. Post-partum hemorrhage (PPH) is a leading cause of death following childbirth although pregnancy associated bleeding can occur at all stages of pregnancy. PPH results in ~100,000 deaths of young and otherwise healthy women each year. Early identification of women at risk for PPH together with appropriate management could reduce the mortality and morbidity associated with PPH. No associations were observed for bleeding in early pregnancy or antepartum hemorrhage. The study identified five genome-wide significant signals associated with PPH. Functional annotation identified HAND2, TBX3, and RAP2C/FRMD7 as likely candidate genes at three loci and showed that associated variants at each locus were located within binding sites for progesterone receptors.

The methods provide general phenotypic definitions for bleeding during pregnancy and PPH sub-phenotype categories defined using hospital admission codes with details provided in Supplementary Table 13. However, the authors also note that not all codes were available in all countries. Detailed descriptions on what information was recorded, how participants were

ascertained and inclusion/exclusion criteria for the individual studies should be provided. What information was available to exclude pregnancies delivered by caesarean section for the PPH analyses and did this vary across the cohorts? How accurate was the ascertainment of cases with retained placenta or uterine atony and did this vary across cohorts? Would variation in ascertainment for the different phenotypes across the different cohorts be likely to influence the results reported?

Answer: The exclusion of pregnancies was only possible in the CHB and MoBa cohorts. The other cohorts were only able to provide a cross-sectional analysis, not taking into account the event in the specific pregnancy (e.g. early bleeding, caesarean section, multifold pregnancy). Case ascertainment is difficult to answer, as this has not been investigated at the national nor cohort levels. In the Danish cohort, it has since 2012 been mandatory to report the amount of blood lost during pregnancy. This is not the case for the other cohorts. We have added the following to clarify:

“Exclusion of multifold pregnancies and delivery by cesarean section were only applicable in the CHB and MoBa cohorts.”

Variation in ascertainment between cohorts could inflate the genetic correlation between the subtypes. However, we have taken great care in computing this. We have iteratively compared the sub-types two cohorts at a time, and then meta analyzed the results. Of course, this was not possible with the phenotype “early bleeding – birth”, due to the small sample size in other cohorts.

The authors report enrichment of PPH signals in PGR binding sites in human embryonic stem cells and nominal significance in the myometrium. The role of dysregulation in the myometrium is highlighted in the title and abstract. What other evidence is there for a role of these genes in regulation in myometrium affecting PPH?

Answer: The evidence we find for the association between PPH and the myometrium is, indeed, at a nominal level. We have adjusted the title and abstract to focus on the progesterone signaling regulation. Furthermore, we have tried to detail the involvement of the five identified genes in regulation of the myometrium in the Discussion, albeit this remains speculative without functional studies:

“The presence of progesterone-binding sites suggests that genes in these regions, such as HAND2, which influences uterine development, PHACTR2, involved in actin regulation for muscle contraction, ZEB1, pivotal in tissue remodeling, TBX3, associated with developmental processes in uterine function, and RAP2C, a mediator of cellular dynamics, may play roles in regulating myometrial contractility”

What is known about the role of progesterone and PGR in the regulation of reported candidate genes? Progesterone is important in the establishment and maintenance of pregnancy. Could factors during the establishment of pregnancy influence the frequency of PPH or is the role of these genes restricted to effects around parturition?

Answer: This is an extremely good point, which we have been discussing for a long time in the author group. The exact timing of these genes is unknown. As the reviewer is aware, there is a scarcity of pregnancy related tissue available for these kinds of analyses. Sampling human tissue *in situ* is not possible, and the placental expression data available comes from at-term pregnancies, typically without complications. We have added this to the Discussion:

*“Nonetheless, due to the lack of relevant *in situ* tissue investigating the exact timing of these genes are unknown”*

The authors report that some PPH variants were also associated with endometriosis and/or uterine fibroids and an inverse genetic correlation between PPH and uterine fibroids. What is the significance of the highlighted genes and reported overlaps between PPH and endometriosis, and the inverse correlation uterine fibroids?

Answer: The negative correlation that we observe with uterine fibroids is not obvious. Large uterine fibroids are a risk factor for postpartum hemorrhage. There may be multiple reasons as to why we observe a negative correlation, including both biological and environmental. If a fibroma is known, obstetric management may change. For instance, a uterine fibroma may prevent vaginal birth, and a cesarean section is performed instead, which lowers the risk of postpartum hemorrhage. *In vitro* studies have also found that progesterone promotes fibroid cell survival.

Due to the significant genetic correlation with uterine fibroids and endometriosis, it is probable that there are shared regions within the genome, yet this hypothesis necessitates validation through analysis with a substantially larger sample size.

The PRS provided a marginal improvement in prediction for PPH. Is family history a risk factor? What increase in power for genetic studies will be required before PRS prediction for PPH is clinically relevant?

Answer: The application of PRS for prediction in a clinical setting depends on multiple aspects. As we also emphasized in the Discussion, there are competing outcomes, such as the possibility of a cesarean section, and preventive measures, such as the use of prophylactic treatments like oxytocin. Nonetheless, there is a marginal improvement, which will only be improved with larger and more diverse cohorts. Family history is indeed a risk factor, given that there is a genetic pre-disposition (see ref 4), albeit the predictive value has not been assessed. Whether there are shared environmental effects remain to be answered.

In the second paragraph of the results referring to the associations with endometriosis and/or uterine fibroids (the text refers readers to Table 2, but this should be Table 1).

Answer: Thank you, that is correct. This has been corrected.

The legend to Table 1 states that endometriosis and uterine fibroids estimates come from the datasets listed in Supplementary Table 4. However, Supplementary Table 4 just has a general reference to the GWAS Catalog. What datasets were used for the endometriosis and uterine fibroids estimates in Table 1?

Answer: We have added the corresponding GWAS Catalog accession id.

Reviewer #3:

Remarks to the Author:

Perinatal bleeding, including early gestation and postpartum hemorrhage (PPH), is an important public health problem associated with maternal death and many co-morbidities. In a genome-wide association study meta-analysis of cohorts involving European ancestry individuals,

Westergaard et al found five loci associated with PPH, and demonstrated that the effect of the loci is more likely to be via the maternal rather than the fetal genome. No associations were detected for early pregnancy bleeding. Ancillary work showed presence of genetic correlations of PPH and early gestational bleeding with other disorders and traits. The prominent discovery is the observation from annotations of the associated nearby genes and downstream analysis pointing to PGR-responsiveness pathways in PPH, partly supporting previously known signals while providing the importance of maternal genetic variation in these pathways and bleeding complications. Comments/suggestions are listed below.

1. The genes linked to PPH support a known role of progesterone/hormones in pregnancy maintenance and bleeding complications, and provide new evidence for role of maternal genetic variation. The added value of the work would be strengthened with further functional experiment in relevant models (e.g., cell lines in relevant tissues for a more mechanistic understanding of the tissue/cell type, and regulatory elements impacted). This can shed light on mechanisms of the genetic effect, given previous work establishing the PGR-receptor signals in maternal-fetal hormonal/immunological crosstalk and related pathologies.

Answer: The complexity of the maternal-fetal interface presents unique challenges. While in vitro models and organ-on-a-chip technologies offer promising avenues, they are still in developmental stages and may not fully recapitulate the in vivo conditions necessary for accurate modeling of the genetic effects in question. We fully recognize the added impact, however, they are outside the scope of the current study. We believe that our findings have laid groundwork for future studies that we, with more mature model systems, can utilize to build on our genetic insights and translate them into a deeper mechanistic understanding.

2. The heritability estimates do not dissociate maternal and fetal contributions. How much do maternal and fetal genotypes contribute to heritability?

Answer: The cohorts included in this study were not powered to investigate fetal effects, and thus we did not include them. A Swedish observational study investigated fetal contributions to postpartum hemorrhage but could not identify a significant fetal effect (ref 4). We have added this to the Discussion:

“The effect of the identified loci was mediated primarily through the maternal genome. This is in line with a prior observational study that could not detect any fetal contribution to the heritability of severe postpartum hemorrhage (>1000 mL)⁴. Nonetheless, we cannot rule out fetal effects completely, as the cohorts with fetal genetic data available were not well powered.”

3. How much do the maternal genetic effects of the five loci identified explain the variation in PPH? Is the variance explained strong enough to consider PRS of PPH in prediction?

Answer: The polygenic risk score that we present does not rely on the variance explained by the five loci. Rather, the PRS is calculated as the contribution (adjusted for LD) of common variants across the autosomes.

In our study, we have evaluated the improvement of the polygenic risk score, in combination with birth weight. We did observe an improvement in the variance explained, going from 3.2% to 3.8%. As we write in the Discussion, this is not quite yet ready for clinical applications, but it will become relevant for stratification as the sample size and power of the genetic studies increase.

4. The PRS finding lumps genetic effects on birthweight with PPH, making it difficult to know whether the improvement in prediction is due to PRS of birthweight, PRS of PPH, or both. More importantly, what was the improvement in prediction of PPH when only PRS of PPH (but not PRS of birthweight) was included? Was the PRS of birthweight maternal or fetal? Why was PRS of birthweight selected but not PRS of other traits (e.g. gestational duration, fibroids which are significantly genetically correlated with PPH)?

Answer: Inclusion of the PRS for PPH alone did improve the variance explained, albeit the improvement was smaller. We did not include gestational duration and uterine fibroids as the existing public summary stats explain only a very small portion of the variance. The polygenic risk score for birth weight was based on the maternal contribution. We have added this to the Results section:

“The variance explained (Nagelkerke R^2) increased from 3.2% (2.7%; 3.8%) to 3.8% (3.4%; 4.5%), yielding a net improvement of 0.7% (0.5%; 0.9%). A model that included only the PRS for PPH explained 3.4% (2.9%; 3.9%), yielding an improvement of 0.2% (0.1%; 0.4%). Similarly, the AUC

increased from 0.60 (0.59; 0.61) to 0.61 (0.60; 0.62), improving marginally (0.008, 0.005; 0.011), when including both PRS. Including only the PRS for PPH resulted in a smaller improvement in the AUC (0.003, 0.001; 0.006)."

5. In the Abstract, the last sentence ("...whereas early bleeding is a complex trait related to underlying health and possibly socioeconomic status.") looks an overinterpretation and less substantiated by data, given the study's heritability estimate of 12.7% for early bleeding, which is just a little smaller than the heritability estimate for PPH and shows the impact of genetic variation on the pathology. The observation of genetic correlations with SES factors as presented in Line 241-242 is not adequate to justify this conclusion.

Answer: The reviewer is correct that our report of "early bleeding is a complex trait related to underlying health and possibly socioeconomic status" is based on the polygenicity findings, as well as the widespread pleiotropic genetic correlations. Genetic correlations can be biased in multiple ways, e.g., overlap in populations or cases. Here, we have taken detail to mitigate these biases by computing the correlation in disjoint populations. We do believe that the findings are important, but we have toned it down slightly by changing the wording to "associated with":

"Our results suggest that postpartum hemorrhage is related to progesterone signaling dysregulation, whereas early bleeding is a complex trait associated with underlying health and possibly socioeconomic status."

6. In the discussion of lack of signals for early bleeding, phenotypic heterogeneity and polygenicity have been mentioned as potential explanation, but which results/evidence substantiate this explanation is unclear.

Answer: The polygenic nature of early bleeding is based on the finding from LD Score Regression (Supplementary Table 3), in which the mean $\chi^2 > 1$, indicating a polygenic signal. The phenotypic heterogeneity is based on clinical evidence, from where we know that early bleeding in pregnancy can be due to (1) implantation bleeding, (2) trauma, (3) pregnancy loss, abnormal products of conception (ectopic pregnancy, molar pregnancy), (4) infections (pelvic inflammatory diseases, sexually transmitted diseases, or bacterial vaginosis), (5) cervical changes, (6) subchorionic haemorrhage, or (7) unexplained causes. The prevalence of each

cause is, to our knowledge, not known, with the exception of pregnancy loss (accounts for approximately 50%). We have added the description of this to the Discussion:

“Early bleeding is a phenotype with high heterogeneity and may be due to implantation bleeding, trauma, pregnancy loss, abnormal products of conception (ectopic pregnancy, molar pregnancy), infections (pelvic inflammatory diseases, sexually transmitted diseases, or bacterial vaginosis), cervical changes, subchorionic hemorrhage, or unexplained causes.”

Decision Letter, first revision:

13th Dec 2023

Dear Dr Nielsen,

Thank you for submitting your revised manuscript "Pregnancy-Associated Bleeding and Genetics: Five Sequence Variants in the Progesterone Signaling Pathway are associated with postpartum hemorrhage" (NG-A63334R). It has now been seen by the original referees and their comments are below. The reviewers find that the paper has improved in revision, and therefore we'll be happy in principle to publish it in Nature Genetics, pending minor revisions to satisfy the referees' final requests and to comply with our editorial and formatting guidelines.

Sincerely,

Safia Danovi
Editor
Nature Genetics

Reviewer #1 (Remarks to the Author):

The authors have responded satisfactorily to all of my previous concerns, and the manuscript is clearer as a result. Just one tiny point: In the legend of Sup Fig 2, "UK, UK Biobank (England); " should be corrected to "UK, UK Biobank (UK); "

Reviewer #2 (Remarks to the Author):

The authors have addressed all comments by the reviewers and revised the manuscript. I have no further comments.

Reviewer #3 (Remarks to the Author):

I thank the authors for the careful revision on the manuscript, and making the best to address my comments.

I still urge the authors to take out the comment on "early bleeding" (see below) in the abstract singling out "early bleeding" as non-genetic. If it is a "complex trait" as mentioned, there should be genetic influences that could have been missed by the study because of the heterogeneity of the phenotype analyzed and the need for larger studies. None of the findings supports "underlying health" or "SES".

"Our results suggest that postpartum hemorrhage is related to progesterone signaling dysregulation, whereas early bleeding is a complex trait associated with underlying health and possibly socioeconomic status."

Author Rebuttal, first revision:

Ad Reviewer #1: We have changed the legend of Sup Fig 2 to "UK, UK Biobank (UK)

Ad Reviewer #3: We understand the reviewers attention, but we do firmly believe that, based on the results presented here, early bleeding does associate with underlying health and socioeconomic status. That is not to say we have identified all of the genomic influences, far from. We merely point to the association we found.

Final Decision Letter:

21st Jun 2024

Dear Dr Nielsen,

I am delighted to say that your manuscript "Genome-wide association meta-analysis identifies five loci associated with postpartum hemorrhage" has been accepted for publication in an upcoming issue of Nature Genetics.

Your paper will be published online after we receive your corrections and will appear in print in the next available issue. You can find out your date of online publication by contacting the Nature Press Office (press@nature.com) after sending your e-proof corrections.

Please note that *Nature Genetics* is a Transformative Journal (TJ). Authors may publish their research with us through the traditional subscription access route or make their paper immediately open access through payment of an article-processing charge (APC). Authors will not be required to make a final decision about access to their article until it has been accepted. Find out more about Transformative Journals

Authors may need to take specific actions to achieve compliance with funder and institutional open access mandates. If your research is supported by a funder that requires

immediate open access (e.g. according to Plan S principles) then you should select the gold OA route, and we will direct you to the compliant route where possible. For authors selecting the subscription publication route, the journal's standard licensing terms will need to be accepted, including <https://www.nature.com/nature-portfolio/editorial-policies/self-archiving-and-license-to-publish>. Those licensing terms will supersede any other terms that the author or any third party may assert apply to any version of the manuscript.

If you have not already done so, we strongly recommend that you upload the step-by-step protocols used in this manuscript to protocols.io. protocols.io is an open online resource that allows researchers to share their detailed experimental know-how. All uploaded protocols are made freely available and are assigned DOIs for ease of citation. Protocols can be linked to any publications in which they are used and will be linked to from your article. You can also establish a dedicated workspace to collect all your lab Protocols. By uploading your Protocols to protocols.io, you are enabling researchers to more readily reproduce or adapt the methodology you use, as well as increasing the visibility of your protocols and papers. Upload your Protocols at <https://protocols.io>. Further information can be found at <https://www.protocols.io/help/publish-articles>.

Sincerely,

Safia Danovi, PhD
Senior Editor, Nature Genetics
ORCID: 0009-0007-7822-5479